# Group Retention when Using Machine Learning in Sequential Decision Making: the Interplay between User Dynamics and Fairness

**Xueru Zhang**[*]
University of Michigan, AnnArbor, USA
xueru@umich.edu

**Mohammad Mahdi Khalili**[*]
University of Michigan, AnnArbor, USA
khalili@umich.edu

**Cem Tekin**
Bilkent University, Ankara, Turkey
cemtekin@ee.bilkent.edu.tr

**Mingyan Liu**
University of Michigan, AnnArbor, USA
mingyan@umich.edu

## Abstract

Machine Learning (ML) models trained on data from multiple demographic groups can inherit representation disparity [7] that may exist in the data: the model may be less favorable to groups contributing less to the training process; this in turn can degrade population retention in these groups over time, and exacerbate representation disparity in the long run. In this study, we seek to understand the interplay between ML decisions and the underlying group representation, how they evolve in a sequential framework, and how the use of fairness criteria plays a role in this process. We show that the representation disparity can easily worsen over time under a natural user dynamics (arrival and departure) model when decisions are made based on a commonly used objective and fairness criteria, resulting in some groups diminishing entirely from the sample pool in the long run. It highlights the fact that fairness criteria have to be defined while taking into consideration the impact of decisions on user dynamics. Toward this end, we explain how a proper fairness criterion can be selected based on a general user dynamics model.

## 1   Introduction

Machine learning models developed from real-world data can inherit pre-existing bias in the dataset. When these models are used to inform decisions involving humans, it may exhibit similar discrimination against sensitive attributes (e.g., gender and race) [6, 14, 15]. Moreover, these decisions can influence human actions, such that bias in the decision is then captured in the dataset used to train future models. This closed feedback loop becomes self-reinforcing and can lead to highly undesirable outcomes over time by allowing biases to perpetuate. For example, speech recognition products such as Amazon's Alexa and Google Home are shown to have accent bias against non-native speakers [6], with native speakers experience much higher quality than non-native speakers. If this difference leads to more native speakers using such products while driving away non-native speakers, then over time the data used to train the model may become even more skewed toward native speakers, with fewer and fewer non-native samples. Without intervention, the resulting model becomes even more accurate for the former and less for the latter, which then reinforces their respective user experience [7].

To address the fairness issues, one commonly used approach is to impose fairness criteria such that certain statistical measures (e.g., positive classification rate, false positive rate, etc.) across different

---

[*]Equal contribution

demographic groups are (approximately) equalized [1]. However, their effectiveness is studied mostly in a static framework, where only the immediate impact of the learning algorithm is assessed but not their long-term consequences. Consider an example where a lender decides whether or not to approve a loan application based on the applicant's credit score. Decisions satisfying an identical true positive rate (equal opportunity) across different racial groups can make the outcome seem fairer [5]. However, this can potentially result in more loans issued to less qualified applicants in the group whose score distribution skews toward higher default risk. The lower repayment among these individuals causes their future credit scores to drop, which moves the score distribution of that group further toward high default risk [13]. This shows that intervention by imposing seemingly fair decisions in the short term can lead to undesirable results in the long run.

In this paper we are particularly interested in understanding what happens to group representation over time when models with fairness guarantee are used, and how it is affected when the underlying feature distributions are also affected/reshaped by decisions. Toward this end, we introduce a user retention model to capture users' reaction (stay or leave) to the decision. We show that under relatively mild and benign conditions, group representation disparity exacerbates over time and eventually the disadvantaged groups may diminish entirely from the system. This condition unfortunately can be easily satisfied when decisions are made based on a typical algorithm (e.g., taking objective as minimizing the total loss) under some commonly used fairness criteria (e.g., statistical parity, equal of opportunity, etc.). Moreover, this exacerbation continues to hold and can accelerate when feature distributions are affected and change over time. A key observation is that if the factors equalized by the fairness criterion do not match what drives user retention, then the difference in (perceived) treatment will exacerbate representation disparity over time. Therefore, fairness has to be defined with a good understanding of how users are affected by the decisions, which can be challenging in practice as we typically have only incomplete/imperfect information. However, we show that if a model for the user dynamics is available, then it is possible to find the proper fairness criterion that mitigates representation disparity.

The impact of fairness intervention on both individuals and society has been studied in [7, 9, 10, 12, 13] and [7, 9, 13] are the most relevant to the present study. Specifically, [9, 13] focus on the impact on reshaping features over two time steps, while we study the impact on group representation over an infinite horizon. [7] studies group representation disparity in a sequential framework but without inspecting the impact of fairness criteria or considering feature distributions reshaped by decision. More on related work can be found in Appendix B.

The remainder of this paper is organized as follows. Section 2 formulates the problem. The impact of various fairness criteria on group representation disparity is analyzed and presented in Section 3, as well as potential mitigation. Experiments are presented in Section 4. Section 5 concludes the paper. All proofs and a table of notations can be found in the appendices.

## 2 Problem Formulation

Consider two demographic groups $G_a$, $G_b$ distinguished based on some sensitive attribute $K \in \{a, b\}$ (e.g., gender, race). An individual from either group has feature $X \in \mathbb{R}^d$ and label $Y \in \{0, 1\}$, both can be time varying. Denote by $G_k^j \subset G_k$ the subgroup with label $j$, $j \in \{0, 1\}$, $k \in \{a, b\}$, $f_{k,t}^j(x)$ its feature distribution and $\alpha_k^j(t)$ the size of $G_k^j$ as a fraction of the entire population at time $t$. Then $\overline{\alpha}_k(t) := \alpha_k^0(t) + \alpha_k^1(t)$ is the size of $G_k$ as a fraction of the population and the difference between $\overline{\alpha}_a(t)$ and $\overline{\alpha}_b(t)$ measures the representation disparity between two groups at time step $t$. Denote by $g_{k,t}^j = \frac{\alpha_k^j(t)}{\overline{\alpha}_k(t)}$ the fraction of label $j \in \{0, 1\}$ in group $k$ at time $t$, then the distribution of $X$ over $G_k$ is given by $f_{k,t}(x) = g_{k,t}^1 f_{k,t}^1(x) + g_{k,t}^0 f_{k,t}^0(x)$ and $f_{a,t} \neq f_{b,t}$.

Consider a sequential setting where the decision maker at each time makes a decision on each individual based on feature $x$. Let $h_\theta(x)$ be the decision rule parameterized by $\theta \in \mathbb{R}^d$ and $\theta_k(t)$ be the decision parameter for $G_k$ at time $t$, $k \in \{a, b\}$. The goal of the decision maker at time $t$ is to find the best parameters $\theta_a(t)$, $\theta_b(t)$ such that the corresponding decisions about individuals from $G_a$, $G_b$ maximize its utility (or minimize its loss) in the current time. Within this context, the commonly studied fair machine learning problem is the one-shot problem stated as follows, at time step $t$:

$$\min_{\theta_a, \theta_b} \boldsymbol{O}_t(\theta_a, \theta_b; \overline{\alpha}_a(t), \overline{\alpha}_b(t)) = \overline{\alpha}_a(t) O_{a,t}(\theta_a) + \overline{\alpha}_b(t) O_{b,t}(\theta_b) \quad \text{s.t.} \quad \Gamma_{\mathcal{C},t}(\theta_a, \theta_b) = 0, \quad (1)$$

where $\boldsymbol{O}_t(\theta_a, \theta_b; \overline{\alpha}_a(t), \overline{\alpha}_b(t))$ is the overall objective of the decision maker at time $t$, which consists of sub-objectives from two groups weighted by their group proportions.[2] $\Gamma_{\mathcal{C},t}(\theta_a, \theta_b) = 0$ charac­terizes fairness constraint $\mathcal{C}$, which requires the parity of certain statistical measure (e.g., positive classification rate, false positive rate, etc.) across different demographic groups. Some commonly used criteria will be elaborated in Section 3.1. Both $O_{k,t}(\theta_k)$ and $\Gamma_{\mathcal{C},t}(\theta_a, \theta_b) = 0$ depend on $f_{k,t}(x)$. The resulting solution $(\theta_a(t), \theta_b(t))$ will be referred to as the one-shot fair decision under fairness $\mathcal{C}$, where the optimality only holds for a single time step $t$.

In this study, we seek to understand how the group representation evolves in a sequential setting over the long run when different fairness criteria are imposed. To do so, the impact of the current decision on the size of the underlying population is modeled by the following discrete-time retention/attrition dynamics. Denote by $N_k(t) \in \mathbb{R}_+$ the expected number of users in group $k$ at time $t$:

$$N_k(t+1) = N_k(t) \cdot \pi_{k,t}(\theta_k(t)) + \beta_k \, , \forall k \in \{a, b\}, \tag{2}$$

where $\pi_{k,t}(\theta_k(t))$ is the retention rate, i.e., the probability of a user from $G_k$ who was in the system at time $t$ remaining in the system at time $t+1$. This is assumed to be a function of the user experience, which could be the actual accuracy of the algorithm or their perceived (mis)treatment. This experience is determined by the application and is different under different contexts. For instance, in domains of speaker verification and medical diagnosis, it can be considered as the average loss, i.e., a user stays if he/she can be classified correctly; in loan/job application scenarios, it can be the rejection rates, i.e., user stays if he/she gets approval. $\beta_k$ is the expected number of exogenous arrivals to $G_k$ and is treated as a constant in our analysis, though our main conclusion holds when this is modeled as a random variable. Accordingly, the relative group representation for time step $t+1$ is updated as $\overline{\alpha}_k(t+1) = \frac{N_k(t+1)}{N_a(t+1)+N_b(t+1)}, \forall k \in \{a, b\}$.

For the remainder of this paper, $\frac{\overline{\alpha}_a(t)}{\overline{\alpha}_b(t)}$ is used to measure the group representation disparity at time $t$. As $\overline{\alpha}_k(t)$ and $f_{k,t}(x)$ change over time, the one-shot problem (1) is also time varying. In the next section, we examine what happens to $\frac{\overline{\alpha}_a(t)}{\overline{\alpha}_b(t)}$ when one-shot fair decisions are applied in each step.

# 3 Analysis of Group Representation Disparity in the Sequential Setting

Below we present results on the monotonic change of $\frac{\overline{\alpha}_a(t)}{\overline{\alpha}_b(t)}$ when applying one-shot fair decisions in each step. It shows that the group representation disparity can worsen over time and may lead to the extinction of one group under a monotonicity condition stated as follows.

**Monotonicity Condition.** *Consider two one-shot problems defined in* (1) *with objectives* $\widehat{\boldsymbol{O}}(\theta_a, \theta_b; \widehat{\overline{\alpha}}_a, \widehat{\overline{\alpha}}_b)$ *and* $\widetilde{\boldsymbol{O}}(\theta_a, \theta_b; \widetilde{\overline{\alpha}}_a, \widetilde{\overline{\alpha}}_b)$ *over distributions* $\widehat{f}_k(x)$, $\widetilde{f}_k(x)$ *respectively. Let* $(\widehat{\theta}_a, \widehat{\theta}_b)$, $(\widetilde{\theta}_a, \widetilde{\theta}_b)$ *be the corresponding fair decisions. We say that two problems* $\widehat{\boldsymbol{O}}$ *and* $\widetilde{\boldsymbol{O}}$ *satisfy the monotonic­ity condition given a dynamic model if for any* $\widehat{\overline{\alpha}}_a + \widehat{\overline{\alpha}}_b = 1$ *and* $\widetilde{\overline{\alpha}}_a + \widetilde{\overline{\alpha}}_b = 1$ *such that* $\frac{\widehat{\overline{\alpha}}_a}{\widehat{\overline{\alpha}}_b} < \frac{\widetilde{\overline{\alpha}}_a}{\widetilde{\overline{\alpha}}_b}$, *the resulting retention rates satisfy* $\widehat{\pi}_a(\widehat{\theta}_a) < \widetilde{\pi}_a(\widetilde{\theta}_a)$ *and* $\widehat{\pi}_b(\widehat{\theta}_b) > \widetilde{\pi}_b(\widetilde{\theta}_b)$.

Note that this condition is defined over two one-shot problems and a given dynamic model. It is not limited to specific families of objective or constraint functions; nor is it limited to one-dimensional features. The only thing that matters is the group proportions within the system and the retention rates determined by the decisions and the dynamics. It characterizes a situation where when one group's representation increases, the decision becomes more in favor of this group and less favorable to the other, so that the retention rate is higher for the favored group and lower for the other.

**Theorem 1.** *[Exacerbation of representation disparity] Consider a sequence of one-shot problems* (1) *with objective* $\boldsymbol{O}_t(\theta_a, \theta_b; \overline{\alpha}_a(t), \overline{\alpha}_b(t))$ *at each time* $t$. *Let* $(\theta_a(t), \theta_b(t))$ *be the corresponding solution and* $\pi_{k,t}(\theta_k(t))$ *be the resulting retention rate of* $G_k$, $k \in \{a, b\}$ *under a dynamic model* (2). *If the initial states satisfy* $\frac{N_a(1)}{N_b(1)} = \frac{\beta_a}{\beta_b}$, $N_k(2) > N_k(1)$,[3] *and one-shot problems in any two consecutive time steps, i.e.,* $\boldsymbol{O}_t$, $\boldsymbol{O}_{t+1}$, *satisfy the monotonicity condition under the given dynamic model, then*

*the following holds. Let $\diamond$ denote either " $<$ " or " $=$ " or " $>$ ", if $\pi_{a,1}(\theta_a(1)) \diamond \pi_{b,1}(\theta_b(1))$, then $\frac{\overline{\alpha}_a(t+1)}{\overline{\alpha}_b(t+1)} \diamond \frac{\overline{\alpha}_a(t)}{\overline{\alpha}_b(t)}$ and $\pi_{a,t+1}(\theta_a(t+1)) \diamond \pi_{a,t}(\theta_a(t)) \diamond \pi_{b,t}(\theta_b(t)) \diamond \pi_{b,t+1}(\theta_b(t+1))$, $\forall t$.*

Theorem 1 says that once a group's proportion starts to change (increase or decrease), it will continue to change in the same direction. This is because under the monotonicity condition, there is a feedback loop between representation disparity and the one-shot decisions: the former drives the latter which results in different user retention rates in the two groups, which then drives future representation.

The monotonicity condition can be satisfied under some commonly used objectives, dynamics and fairness criteria. This is characterized in the following theorem.

**Theorem 2.** *[A case satisfying monotonicity condition] Consider two one-shot problems defined in (1) with objectives $\widehat{\mathbf{O}}(\theta_a, \theta_b; \widehat{\overline{\alpha}}_a, \widehat{\overline{\alpha}}_b) = \widehat{\overline{\alpha}}_a O_a(\theta_a) + \widehat{\overline{\alpha}}_b O_b(\theta_b)$ and $\widehat{\mathbf{O}}(\theta_a, \theta_b; \widetilde{\overline{\alpha}}_a, \widetilde{\overline{\alpha}}_b) = \widetilde{\overline{\alpha}}_a O_a(\theta_a) + \widetilde{\overline{\alpha}}_b O_b(\theta_b)$ over the same distribution $f_k(x)$ with $\widehat{\overline{\alpha}}_a + \widehat{\overline{\alpha}}_b = 1$ and $\widetilde{\overline{\alpha}}_a + \widetilde{\overline{\alpha}}_b = 1$. Let $(\widehat{\theta}_a, \widehat{\theta}_b)$, $(\widetilde{\theta}_a, \widetilde{\theta}_b)$ be the corresponding solutions. Under the condition that $O_k(\widehat{\theta}_k) \neq O_k(\widetilde{\theta}_k)$ for all possible $\widehat{\overline{\alpha}}_k \neq \widetilde{\overline{\alpha}}_k$, if the dynamics satisfy $\pi_k(\theta_k) = h_k(O_k(\theta_k))$ for some decreasing function $h_k(\cdot)$, then $\widetilde{\mathbf{O}}$ and $\widehat{\mathbf{O}}$ satisfy the monotonicity condition.*

The above theorem identifies a class of cases satisfying the monotonicity condition; these are cases where whenever the group proportion changes, the decision will cause the sub-objective function value to change as well, and the sub-objective function value drives user departure.

For the rest of the paper we will focus on the one-dimensional setting. Some of the cases we consider are special cases of Theorem 2 (Sec. 3.2). Others such as the time-varying feature distribution $f_{k,t}(x)$ considered in Sec. 3.3 also satisfy the monotonicity condition but are not captured by Theorem 2.

## 3.1 The one-shot problem

Consider a binary classification problem based on feature $X \in \mathbb{R}$. Let decision rule $h_\theta(x) = \mathbf{1}(x \geq \theta)$ be a threshold policy parameterized by $\theta \in \mathbb{R}$ and $L(y, h_\theta(x)) = \mathbf{1}(y \neq h_\theta(x))$ the 0-1 loss incurred by applying decision $\theta$ on individuals with data $(x, y)$.

The goal of the decision maker at each time is to find a pair $(\theta_a(t), \theta_b(t))$ subject to criterion $\mathcal{C}$ such that the total expected loss is minimized, i.e., $\boldsymbol{O}_t(\theta_a, \theta_b; \overline{\alpha}_a(t), \overline{\alpha}_b(t)) = \overline{\alpha}_a(t) L_{a,t}(\theta_a) + \overline{\alpha}_b(t) L_{b,t}(\theta_b)$, where $L_{k,t}(\theta_k) = g_{k,t}^1 \int_{-\infty}^{\theta_k} f_{k,t}^1(x)dx + g_{k,t}^0 \int_{\theta_k}^{\infty} f_{k,t}^0(x)dx$ is the expected loss $G_k$ experiences at time $t$. Some examples of $\Gamma_{\mathcal{C},t}(\theta_a, \theta_b)$ are as follows and illustrated in Fig. 1.

1. Simple fair (Simple): $\Gamma_{\texttt{Simple},t} = \theta_a - \theta_b$. Imposing this criterion simply means we ensure the same decision parameter is used for both groups.
2. Equal opportunity (EqOpt): $\Gamma_{\texttt{EqOpt},t} = \int_{\theta_a}^{\infty} f_{a,t}^0(x)dx - \int_{\theta_b}^{\infty} f_{b,t}^0(x)dx$. This requires the false positive rate (FPR) be the same for different groups (Fig. 1(c)),[4] i.e., $\Pr(h_{\theta_a}(X) = 1|Y = 0, K = a) = \Pr(h_{\theta_b}(X) = 1|Y = 0, K = b)$.
3. Statistical parity (StatPar): $\Gamma_{\texttt{StatPar},t} = \int_{\theta_a}^{\infty} f_{a,t}(x)dx - \int_{\theta_b}^{\infty} f_{b,t}(x)dx$. This requires different groups be given equal probability of being labelled 1 (Fig. 1(b)), i.e., $\Pr(h_{\theta_a}(X) = 1|K = a) = \Pr(h_{\theta_b}(X) = 1|K = b)$.
4. Equalized loss (EqLos): $\Gamma_{\texttt{EqLos},t} = L_{a,t}(\theta_a) - L_{b,t}(\theta_b)$. This requires that the expected loss across different groups be equal (Fig. 1(d)).

Notice that for Simple, EqOpt and StatPar criteria, the following holds: $\forall t$, $(\theta_a, \theta_b)$, and $(\theta_a', \theta_b')$ that satisfy $\Gamma_{\mathcal{C},t}(\theta_a, \theta_b) = \Gamma_{\mathcal{C},t}(\theta_a', \theta_b') = 0$, we have $\theta_a \geq \theta_a'$ if and only if $\theta_b \geq \theta_b'$.

Some technical assumptions on the feature distributions are in order. We assume $f_{a,t}^0(x), f_{a,t}^1(x), f_{b,t}^0(x), f_{b,t}^1(x)$ have bounded support on $[\underline{a}_t^0, \overline{a}_t^0]$, $[\underline{a}_t^1, \overline{a}_t^1]$, $[\underline{b}_t^0, \overline{b}_t^0]$ and $[\underline{b}_t^1, \overline{b}_t^1]$ respectively, and that $f_{k,t}^1(x)$ and $f_{k,t}^0(x)$ overlap, i.e., $\underline{a}_t^0 < \underline{a}_t^1 < \overline{a}_t^0 < \overline{a}_t^1$ and $\underline{b}_t^0 < \underline{b}_t^1 < \overline{b}_t^0 < \overline{b}_t^1$. The main technical assumption is stated as follows.

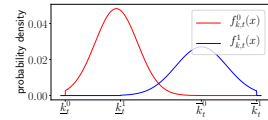

Fig. 2: $f_{k,t}^j(x)$, $k \in \{a, b\}$

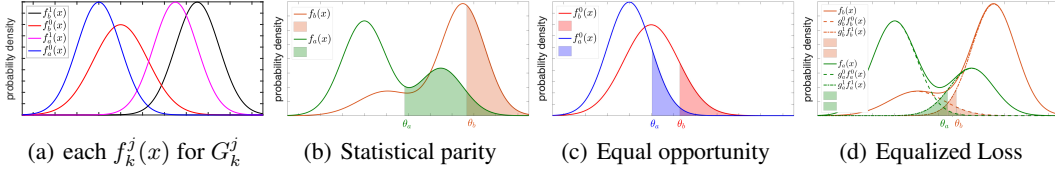

| (a) each $f_k^j(x)$ for $G_k^j$ | (b) Statistical parity | (c) Equal opportunity | (d) Equalized Loss |

Fig. 1: For $G_a$, $G_b$ with group proportions $\alpha_a^1 = 0.55, \alpha_a^0 = 0.15, \alpha_b^1 = 0.1, \alpha_b^0 = 0.2$, a pair of $(\theta_a, \theta_b)$ is fair under each criterion stated in Fig. 1(b)-1(d) requires the corresponding colored areas be equal.

**Assumption 1.** *Let $\mathcal{T}_{a,t} = [\underline{a}_t^1, \overline{a}_t^0]$ (resp. $\mathcal{T}_{b,t} = [\underline{b}_t^1, \overline{b}_t^0]$) be the overlapping interval between $f_{a,t}^0(x)$ and $f_{a,t}^1(x)$ (resp. $f_{b,t}^0(x)$ and $f_{b,t}^1(x)$). Distribution $f_{k,t}^1(x)$ is strictly increasing and $f_{k,t}^0(x)$ is strictly decreasing over $\mathcal{T}_{k,t}$, $\forall k \in \{a, b\}$.*

For bell-shaped feature distributions (e.g., Normal, Cauchy, etc.), Assumption 1 implies that $f_{k,t}^1(x)$ and $f_{k,t}^0(x)$ are sufficiently separated. An example is shown in Fig. 2. As we show later, this assumption helps us establish the monotonic convergence of decisions $(\theta_a(t), \theta_b(t))$ but is not necessary for the convergence of group representation. We next find the one-shot decision to this problem under `Simple`, `EqOpt`, and `StatPar` fairness criteria.

**Lemma 1.** *Under Assumption 1, $\forall k \in \{a, b\}$, the optimal decision at time $t$ for $G_k$ without considering fairness is*

$$\theta_k^*(t) = \arg\min_{\theta_k} L_{k,t}(\theta_k) = \begin{cases} \underline{k}_t^1, & \text{if } g_{k,t}^1 f_{k,t}^1(\underline{k}_t^1) \geq g_{k,t}^0 f_{k,t}^0(\underline{k}_t^1) \\ \delta_{k,t}, & \text{if } g_{k,t}^1 f_{k,t}^1(\underline{k}_t^1) < g_{k,t}^0 f_{k,t}^0(\underline{k}_t^1) \ \& \ g_{k,t}^1 f_{k,t}^1(\overline{k}_t^0) > g_{k,t}^0 f_{k,t}^0(\overline{k}_t^0) \\ \overline{k}_t^0, & \text{if } g_{k,t}^1 f_{k,t}^1(\overline{k}_t^0) \leq g_{k,t}^0 f_{k,t}^0(\overline{k}_t^0) \end{cases}$$

*where $\delta_{k,t} \in \mathcal{T}_{k,t}$ is defined such that $g_{k,t}^1 f_{k,t}^1(\delta_{k,t}) = g_{k,t}^0 f_{k,t}^0(\delta_{k,t})$. Moreover, $L_{k,t}(\theta_k)$ is decreasing in $\theta_k$ over $[\underline{k}_t^0, \theta_k^*(t)]$ and increasing over $[\theta_k^*(t), \overline{k}_t^1]$.*

Below we will focus on the case when $\theta_a^*(t) = \delta_{a,t}$ and $\theta_b^*(t) = \delta_{b,t}$, while analysis for the other cases are essentially the same. For `Simple`, `StatPar` and `EqOpt` fairness, $\exists$ a strictly increasing function $\phi_{\mathcal{C},t}$, such that $\Gamma_{\mathcal{C},t}(\phi_{\mathcal{C},t}(\theta_b), \theta_b) = 0$. Denote by $\phi_{\mathcal{C},t}^{-1}$ the inverse of $\phi_{\mathcal{C},t}$. Without loss of generality, we will assign group labels $a$ and $b$ such that $\phi_{\mathcal{C},t}(\delta_{b,t}) < \delta_{a,t}$ and $\phi_{\mathcal{C},t}^{-1}(\delta_{a,t}) > \delta_{b,t}, \forall t$. [5]

**Lemma 2.** *Under `Simple`, `EqOpt`, `StatPar` fairness criteria, one-shot fair decision at time $t$ satisfies $(\theta_a^*(t), \theta_b^*(t)) = \arg\min_{\theta_a, \theta_b} \overline{\alpha}_a(t) L_{a,t}(\theta_a) + \overline{\alpha}_b(t) L_{b,t}(\theta_b) \in \{(\theta_a, \theta_b) | \theta_a \in [\phi_{\mathcal{C},t}(\delta_{b,t}), \delta_{a,t}], \theta_b \in [\delta_{b,t}, \phi_{\mathcal{C},t}^{-1}(\delta_{a,t})], \Gamma_{\mathcal{C},t}(\theta_a, \theta_b) = 0\} \neq \emptyset$ regardless of group proportions $\overline{\alpha}_a(t), \overline{\alpha}_b(t)$.*

Lemma 2 shows that given feature distributions $f_{a,t}(x), f_{b,t}(x)$, although one-shot fair decisions can be different under different group proportions $\overline{\alpha}_a(t), \overline{\alpha}_b(t)$, these solutions are all bounded by the same compact intervals (Fig. 3). Theorem 3 below describes the more specific relationship between group representation $\frac{\overline{\alpha}_a(t)}{\overline{\alpha}_b(t)}$ and the corresponding one-shot decision $(\theta_a(t), \theta_b(t))$.

**Theorem 3.** *[Impact of group representation disparity on the one-shot decision] Consider the one-shot problem with group proportions $\overline{\alpha}_a(t), \overline{\alpha}_b(t)$ at time step $t$, let $(\theta_a(t), \theta_b(t))$ be the corresponding one-shot decision under either `Simple`, `EqOpt` or `StatPar` criterion. Under Assumption 1, $(\theta_a(t), \theta_b(t))$ is unique and satisfies the following:*

$$\Psi_{\mathcal{C},t}(\theta_a(t), \theta_b(t)) = \frac{\overline{\alpha}_a(t)}{\overline{\alpha}_b(t)}, \tag{3}$$

*where $\Psi_{\mathcal{C},t}$ is some function increasing in $\theta_a(t)$ and $\theta_b(t)$, with details illustrated in Table 1.*

| | $\theta_a \in [\underline{a}_t^0, \underline{a}_t^1], \theta_b \in \mathcal{T}_{b,t}$ | $\theta_a \in \mathcal{T}_{a,t}, \theta_b \in \mathcal{T}_{b,t}$ | $\theta_a \in \mathcal{T}_{a,t}, \theta_b \in [\bar{b}_t^0, \bar{b}_t^1]$ |
|---|---|---|---|
| EqOpt | $\left(\dfrac{g_{b,t}^1 \, f_{b,t}^1(\theta_b)}{g_{b,t}^0 \, f_{b,t}^0(\theta_b)} - 1\right)\dfrac{g_{b,t}^0}{g_{a,t}^0}$ | $\dfrac{\frac{g_{b,t}^1 \, f_{b,t}^1(\theta_b)}{g_{b,t}^0 \, f_{b,t}^0(\theta_b)} - 1}{1 - \frac{g_{a,t}^1 \, f_{a,t}^1(\theta_a)}{g_{a,t}^0 \, f_{a,t}^0(\theta_a)}}\dfrac{g_{b,t}^0}{g_{a,t}^0}$ | |
| StatPar | $1 - \dfrac{2}{\frac{g_{b,t}^1 \, f_{b,t}^1(\theta_b)}{g_{b,t}^0 \, f_{b,t}^0(\theta_b)} + 1}$ | $\left(1 - \dfrac{2}{\frac{g_{b,t}^1 \, f_{b,t}^1(\theta_b)}{g_{b,t}^0 \, f_{b,t}^0(\theta_b)} + 1}\right)\left(\dfrac{2}{1 - \frac{g_{a,t}^1 \, f_{a,t}^1(\theta_a)}{g_{a,t}^0 \, f_{a,t}^0(\theta_a)}} - 1\right)$ | $\dfrac{2}{1 - \frac{g_{a,t}^1 \, f_{a,t}^1(\theta_a)}{g_{a,t}^0 \, f_{a,t}^0(\theta_a)}} - 1$ |
| Simple | | $\dfrac{g_{b,t}^1 f_{b,t}^1(\theta_b) - g_{b,t}^0 f_{b,t}^0(\theta_b)}{g_{a,t}^0 f_{a,t}^0(\theta_a) - g_{a,t}^1 f_{a,t}^1(\theta_a)}$ | |

Table 1: The form of $\Psi_{\mathcal{C},t}(\theta_a, \theta_b)$ for $\mathcal{C} = $ EqOpt, StatPar, Simple.[6]

Note that under Assumption 1, both $\frac{g_{k,t}^1 f_{k,t}^1(\theta_k)}{g_{k,t}^0 f_{k,t}^0(\theta_k)}$ and $g_{k,t}^1 f_{k,t}^1(\theta_k) - g_{k,t}^0 f_{k,t}^0(\theta_k)$ are strictly increasing in $\theta_k \in \mathcal{T}_{k,t}$, $k \in \{a, b\}$, and $\theta_a(t) = \phi_{\mathcal{C},t}(\theta_b(t))$ for some strictly increasing function. According to $\Psi_{\mathcal{C},t}(\theta_a, \theta_b)$ given in Table 1, the larger $\frac{\overline{\alpha}_a(t)}{\overline{\alpha}_b(t)}$ results in the larger $\frac{g_{k,t}^1 f_{k,t}^1(\theta_k)}{g_{k,t}^0 f_{k,t}^0(\theta_k)}$ and $g_{k,t}^1 f_{k,t}^1(\theta_k) - g_{k,t}^0 f_{k,t}^0(\theta_k)$, thus the larger $\theta_a(t)$ and $\theta_b(t)$. The above theorem characterizes the impact of the underlying population on the one-shot decisions. Next we investigate how the one-shot decision impacts the underlying population.

### 3.2 Participation dynamics

How a user reacts to the decision is captured by the retention dynamics (2) which is fully characterized by the retention rate. Below we introduce two types of (perceived) mistreatment as examples when the monotonicity condition is satisfied.

**(1) User departure driven by model accuracy:** Examples include discontinuing the use of products viewed as error-prone, e.g., speech recognition software, or medical diagnostic tools. In these cases, the determining factor is the classification error, i.e., users who experience low accuracy have a higher probability of leaving the system. The retention rate at time $t$ can be modeled as $\pi_{k,t}(\theta_k) = \nu(L_{k,t}(\theta_k))$ for some strictly *decreasing* function $\nu(\cdot) : [0, 1] \to [0, 1]$.

**(2) User departure driven by intra-group disparity:** Participation can also be affected by intra-group disparity, that between users from the same demographic group but with different labels, i.e., $G_k^j$ for $j \in \{0, 1\}$. An example is in making financial assistance decisions where one expects to see more awards given to those qualified than to those unqualified. Denote by $D_{k,t}(\theta_k) = \Pr(Y = 1, h_{\theta_k}(X) = 1|K = k) - \Pr(Y = 0, h_{\theta_k}(X) = 1|K = k) = \int_{\theta_k}^{\infty} \left(g_k^1 f_{k,t}^1(x) - g_k^0 f_{k,t}^0(x)\right)dx$ as intra-group disparity of $G_k$ at time $t$, then the retention rate can be modeled as $\pi_{k,t}(\theta_k) = w(D_{k,t}(\theta_k))$ for some strictly *increasing* function $w(\cdot)$ mapping to $[0, 1]$.

**Theorem 4.** *Consider the one-shot problem* (1) *defined in Sec. 3.1 under either* Simple, EqOpt *or* StatPar *criterion, and assume distributions* $f_{k,t}(x) = f_k(x)$ *are fixed over time. Then the one-shot problems in any two consecutive time steps, i.e.,* $\mathbf{O}_t, \mathbf{O}_{t+1}$, *satisfy the monotonicity condition under dynamics* (2) *with* $\pi_k(\cdot)$ *being either* $\nu(L_k(\cdot))$ *or* $w(D_k(\cdot))$.[7] *This implies that Theorem 1 holds and* $(\theta_a(t), \theta_b(t))$ *converges monotonically to a constant decision* $(\theta_a^\infty, \theta_b^\infty)$. *Furthermore,* $\lim_{t \to \infty} \frac{\overline{\alpha}_a(t)}{\overline{\alpha}_b(t)} = \frac{\beta_a}{\beta_b} \frac{1 - \pi_b(\theta_b^\infty)}{1 - \pi_a(\theta_a^\infty)}$.

When distributions are fixed, the discrepancy between $\pi_a(\theta_a(t))$ and $\pi_b(\theta_b(t))$ increases over time as $(\theta_a(t), \theta_b(t))$ changes. The process is illustrated in Fig. 3, where $\theta_a(t) \in [\phi_{\mathcal{C}}(\delta_b), \delta_a], \theta_b(t) \in [\delta_b, \phi_{\mathcal{C}}^{-1}(\delta_a)]$ are constrained by the same interval $\forall t$. Left and right plots illustrate cases when $\pi_k(\theta_k) = \nu(L_k(\theta_k))$ and $\pi_k(\theta_k) = w(D_k(\theta_k))$ respectively.

Note that the case considered in Theorem 4 is a special case of Theorem 2, with distributions $f_{k,t}(x) = f_k(x)$ fixed, $O_k(\theta_k) = L_k(\theta_k)$ and both dynamics $\pi_k(\cdot) = \nu(L_k(\cdot))$ and $\pi_k(\cdot) = w(D_k(\cdot))$ some

decreasing functions of $L_k(\cdot)$.[8] In this special case we obtain the additional result of monotonic convergence of decisions, which holds due to Assumption 1.

Once $\frac{\overline{\alpha}_a(t)}{\overline{\alpha}_b(t)}$ starts to increase, the corresponding one-shot solution $(\theta_a(t), \theta_b(t))$ also increases (Theorem 3), meaning that $\theta_a(t)$ moves closer to $\theta_a^* = \delta_a$ and $\theta_b(t)$ moves further away from $\theta_b^* = \delta_b$ (solid arrows in Fig. 3). Consequently, $L_a(\theta_a(t))$ and $D_b(\theta_b(t))$ decrease while $L_b(\theta_b(t))$ and $D_a(\theta_a(t))$ increase. Under both dynamics, $\pi_a(\theta_a(t))$ increases and $\pi_b(\theta_b(t))$ decreases, resulting in the increase of $\frac{\overline{\alpha}_a(t+1)}{\overline{\alpha}_b(t+1)}$; the feedback loop becomes self-reinforcing and representation disparity worsens.

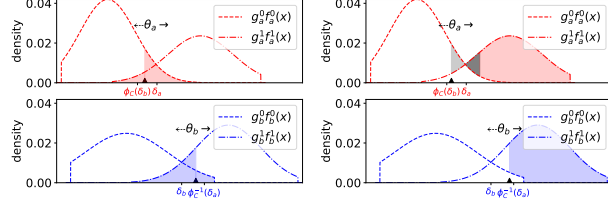

Fig. 3: Illustration of $L_k(\theta_k)$ and $D_k(\theta_k)$ w.r.t. $\theta_k$: Each black triangle represents the one-shot decision $\theta_k$; size of the colored area represents the value of $L_k(\theta_k)$ (left) or $D_k(\theta_k)$ (right). Note that for the right plot, there are two gray regions and the darker one is for compensating the lighter one thus they are of the same size; the smaller gray regions result in the larger $D_a(\theta_a)$.

### 3.3 Impact of decisions on reshaping feature distributions

Our results so far show the potential adverse impact on group representation when imposing certain fairness criterion, while their underlying feature distributions are assumed fixed. Below we examine what happens when decisions also affect feature distributions over time, i.e., $f_{k,t}(x) = g_{k,t}^1 f_{k,t}^1(x) + g_{k,t}^0 f_{k,t}^0(x)$, which is not captured by Theorem 2. We will focus on the dynamics $\pi_{k,t}(\theta_k) = \nu(L_{k,t}(\theta_k))$. Since $G_k^0, G_k^1$ may react differently to the same $\theta_k$, we consider two scenarios as illustrated in Fig. 4, which shows the change in distribution from $t$ to $t+1$ when $G_k^1$ (resp. $G_k^0$) experiences the higher (resp. lower) loss at $t$ than $t-1$ (see Appendix I for more detail): $\forall j \in \{0, 1\}$,

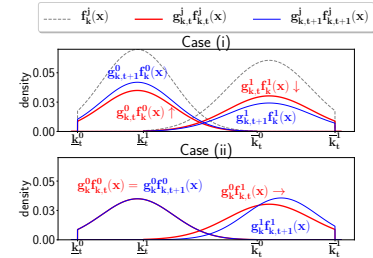

Fig. 4: Visualization of decisions shaping feature distributions.

**Case (i):** $f_{k,t}^j(x) = f_k^j(x)$ remain fixed but $g_{k,t}^j$ changes over time given $G_k^j$'s retention determined by its perceived loss $L_{k,t}^j$,[9] In other words, for $i \in \{0, 1\}$ and $t \geq 2$ such that $L_{k,t}^i(\theta_k(t)) < L_{k,t-1}^i(\theta_k(t-1))$, we have $g_{k,t+1}^i > g_{k,t}^i$ and $g_{k,t+1}^{-i} < g_{k,t}^{-i}$, where $-i := \{0, 1\} \setminus \{i\}$.

**Case (ii):** $g_{k,t}^j = g_k^j$ but for subgroup $G_k^i$ that is less favored by the decision over time, its members make extra effort such that $f_{k,t}^i(x)$ skews toward the direction of lowering their losses.[10] In other words, for $i \in \{0, 1\}$ and $t \geq 2$ such that $L_{k,t}^i(\theta_k(t)) > L_{k,t-1}^i(\theta_k(t-1))$, we have $f_{k,t+1}^i(x) < f_{k,t}^i(x)$, $\forall x \in \mathcal{T}_k$, while $f_{k,t+1}^{-i}(x) = f_{k,t}^{-i}(x)$, $\forall x$, where $-i := \{0, 1\} \setminus \{i\}$.

In both cases, under the condition that $f_{k,t}(x)$ is relatively insensitive to the change in one-shot decisions, representation disparity can worsen and deterioration accelerates. The precise conditions are formally given in Conditions 1 and 2 in Appendix I, which describes the case where the change from $f_{k,t}(x)$ to $f_{k,t+1}(x)$ is sufficiently small while the change from $\frac{\overline{\alpha}_a(t)}{\overline{\alpha}_b(t)}$ to $\frac{\overline{\alpha}_a(t+1)}{\overline{\alpha}_b(t+1)}$ and the resulting decisions from $\theta_k(t)$ to $\theta_k(t+1)$ are sufficiently large. These conditions hold in scenarios when the change in feature distributions induced by the one-shot decisions is a slow process.

**Theorem 5.** *[Exacerbation in representation disparity can accelerate] Consider the one-shot problem defined in* (1) *under either* `Simple`*,* `EqOpt` *or* `StatPar` *fairness criterion. Let the one-shot decision, representation disparity and retention rate at time $t$ be given by $\theta_k^f(t)$, $\frac{\overline{\alpha}_a^f(t)}{\overline{\alpha}_b^f(t)}$, and $\pi_{k,t}^f(\theta_k^f(t))$ when distribution $f_k(x)$ is fixed $\forall t$. Let the same be denoted by $\theta_k^r(t)$, $\frac{\overline{\alpha}_a^r(t)}{\overline{\alpha}_b^r(t)}$, and $\pi_{k,t}^r(\theta_k^r(t))$ when $f_{k,t}(x)$ changes according to either case (i) or (ii) defined above. Assume we start from the*

*same distribution $f_{k,1}(x) = f_k(x)$. Under Conditions 1 and 2 in Appendix I, if $\pi_{a,1}^f(\theta_a^f(1)) = \pi_{a,1}^r(\theta_a^r(1)) \diamond \pi_{b,1}^f(\theta_b^f(1)) = \pi_{b,1}^r(\theta_b^r(1))$, then $\frac{\overline{\alpha}_a^r(t+1)}{\overline{\alpha}_b^r(t+1)} \diamond \frac{\overline{\alpha}_a^r(t)}{\overline{\alpha}_b^r(t)}$ (disparity worsens) and $\frac{\overline{\alpha}_a^r(t+1)}{\overline{\alpha}_b^r(t+1)} \diamond \frac{\overline{\alpha}_a^f(t+1)}{\overline{\alpha}_b^f(t+1)}$ (accelerates), $\forall t$, where $\diamond$ represents either " < " or " > ".*

### 3.4 Potential mitigation & finding the proper fairness criterion from participation dynamics

The above results show that when the objective is to minimize the average loss over the entire population, applying commonly used and seemingly fair decisions at each time can exacerbate representation disparity over time under reasonable participation dynamics. It highlights the fact that fairness has to be defined with a good understanding of how users are affected by the algorithm, and how they may react to it. For instance, consider the dynamics with $\pi_{k,t}(\theta_k) = \nu(L_{k,t}(\theta_k))$, then imposing EqLos fairness (Fig. 1(d)) at each time step would sustain group representations, i.e., $\lim_{t\to\infty}\frac{\overline{\alpha}_a(t)}{\overline{\alpha}_b(t)} = \frac{\beta_a}{\beta_b}$, as we are essentially equalizing departure when equalizing loss. In contrast, under other fairness criteria the factors that are equalized do not match what drives departure, and different losses incurred to different groups cause significant change in group representation over time.

In reality the true dynamics is likely a function of a mixture of factors given the application context, and a proper fairness constraint $\mathcal{C}$ should be adopted accordingly. Below we illustrate a method for finding the proper criterion from a general dynamics model defined below when $f_{k,t}(x) = f_k(x), \forall t$:

$$N_k(t+1) = \Lambda(N_k(t), \{\pi_k^m(\theta_k(t))\}_{m=1}^M, \beta_k), \ \forall k \in \{a,b\}, \tag{4}$$

where user retention in $G_k$ is driven by $M$ different factors $\{\pi_k^m(\theta_k(t))\}_{m=1}^M$ (e.g. accuracy, true positives, etc.) and each of them depends on decision $\theta_k(t)$. Constant $\beta_k$ is the intrinsic growth rate while the actual arrivals may depend on $\pi_k^m(\theta_k(t))$. The expected number of users at time $t+1$ depends on users at $t$ and new users; both may be effected by $\pi_k^m(\theta_k(t))$. This relationship is characterized by a general function $\Lambda$. Let $\Theta$ be the set of all possible decisions.

**Assumption 2.** $\exists(\theta_a, \theta_b) \in \Theta \times \Theta$ such that $\forall k \in \{a,b\}$, $\hat{N}_k = \Lambda(\hat{N}_k, \{\pi_k^m(\theta_k)\}_{m=1}^M, \beta_k)$ and $|\Lambda'(\hat{N}_k, \{\pi_k^m(\theta_k)\}_{m=1}^M, \beta_k)| < 1$ hold for some $\hat{N}_k$, i.e., dynamics (4) under some decision pairs $(\theta_a, \theta_b)$ have stable fixed points, where $\Lambda'$ denotes the derivative of $\Lambda$ with respect to $N_k$.

To find the proper fairness constraint, let $\mathcal{C}$ be the set of decisions $(\theta_a, \theta_b)$ that can sustain group representation. It can be found via the following optimization problem; the set of feasible solutions is guaranteed to be non-empty under Assumption 2.

$$\mathcal{C} = \arg\min_{(\theta_a,\theta_b)} \left|\frac{\tilde{N}_a}{\tilde{N}_b} - \frac{\beta_a}{\beta_b}\right| \ \text{s.t.} \ \tilde{N}_k = \Lambda(\tilde{N}_k, \{\pi_k^m(\theta_k)\}_{m=1}^M, \beta_k) \in \mathbb{R}_+, \theta_k \in \Theta, \forall k \in \{a,b\}.$$

The idea is to first select decision pairs whose corresponding dynamics can lead to stable fixed points $(\tilde{N}_a, \tilde{N}_b)$; then among them select those that are best in sustaining group representation, which may or may not be unique. Sometimes guaranteeing the perfect fairness can be unrealistic and a relaxed version is preferred, in which case all pairs $(\theta_a, \theta_b)$ satisfying $|\frac{\tilde{N}_a}{\tilde{N}_b} - \frac{\beta_a}{\beta_b}| \leq \min\{|\frac{\tilde{N}_a}{\tilde{N}_b} - \frac{\beta_a}{\beta_b}|\} + \Delta$ constitute the $\Delta$-fair set. An example under dynamics $N_k(t+1) = N_k(t)\pi_k^2(\theta_k(t)) + \beta_k \pi_k^1(\theta_k(t))$ is illustrated in Fig. 5, where all curves with $\epsilon \leq \Delta\frac{\beta_b}{\beta_a}$ constitute $\Delta$-fair set (perfect fairness set is given by the deepest red curve with $\epsilon = 0$). See Appendix K for more details.

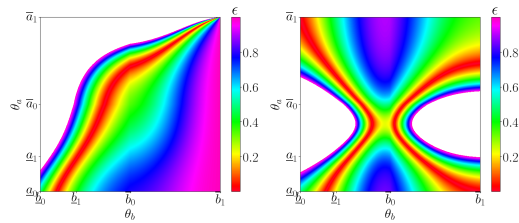

Fig. 5: Left plot: $\pi_k^2(\theta_k) = \nu(\int_{\theta_k}^\infty f_k(x)dx), \pi_k^1(\theta_k) = \nu(L_k(\theta_k))$; right plot: $\pi_k^2(\theta_k) = \nu(L_k(\theta_k)), \pi_k^1(\theta_k) = 1$, and $\nu(x) = 1 - x$. Value of each pair $(\theta_a, \theta_b)$ corresponds to $|\frac{\tilde{N}_a}{\tilde{N}_b} - \frac{\beta_a}{\beta_b}|$ measuring how well it can sustain the group representation. All points $(\theta_a, \theta_b)$ with the same value of $|\frac{\tilde{N}_a}{\tilde{N}_b} - \frac{\beta_a}{\beta_b}| = \frac{\beta_a}{\beta_b}\epsilon$ form a curve of the same color with $\epsilon \in [0,1]$ shown in the color bar.

## 4 Experiments

We first performed a set of experiments on synthetic data where every $G_k^j, k \in \{a,b\}, j \in \{0,1\}$ follows the truncated normal (Fig. 2) distributions. A sequence of one-shot fair decisions are used

and group representation changes over time according to dynamics (2) with $\pi_k(\theta_k) = \nu(L_k(\theta_k))$. Parameter settings and more experimental results (e.g., sample paths, results under other dynamics and when feature distributions are learned from data) are presented in Appendix L.

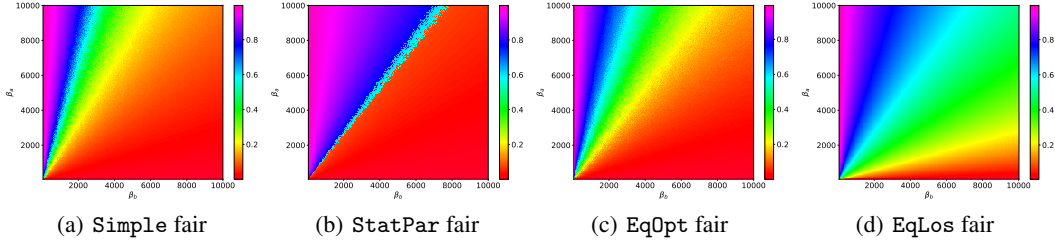

(a) `Simple` fair      (b) `StatPar` fair      (c) `EqOpt` fair      (d) `EqLos` fair

Fig. 6: Each dot in Fig. 6(a)-6(d) represents the final group proportion $\lim_{t\to\infty} \overline{\alpha}_a(t)$ of one sample path under a pair of arriving rates $(\beta_a, \beta_b)$. If the group representation is sustained, then $\lim_{t\to\infty} \overline{\alpha}_a(t) = \frac{1}{1+\beta_b/\beta_a}$ for each pair of $(\beta_a, \beta_b)$, as shown in Fig. 6(d) under `EqLos` fairness. However, under `Simple`, `StatPar` and `EqOpt` fairness, $\lim_{t\to\infty} \overline{\alpha}_a(t) = 1/(1 + \frac{\beta_b(1-\nu(L_a(\theta_a^\infty)))}{\beta_a(1-\nu(L_b(\theta_b^\infty)))})$.

Fig. 6 illustrates the final group proportion (the converged state) $\lim_{t\to\infty} \overline{\alpha}_a(t)$ as a function of the exogenous arrival sizes $\beta_a$ and $\beta_b$ under different fairness criteria. With the exception of `EqLos`

fairness, group representation is severely skewed in the long run, with the system consisting mostly of $G_b$, even for scenarios when $G_a$ has larger arrival, i.e., $\beta_a > \beta_b$. Moreover, decisions under an inappropriate fairness criterion (`Simple`, `EqOpt` or `StatPar`) can result in poor robustness, where a minor change in $\beta_a$ and $\beta_b$ can result in very different representation in the long run (Fig. 6(b)).

We also consider the dynamics presented in Fig. 5 and show the effect of $\Delta = \epsilon \frac{\beta_a}{\beta_b}$-fair decision found with method in Sec. 3.4 on $\overline{\alpha}_a(t)$. Each curve in Fig. 7 represents a sample path under different $\epsilon$ where $(\theta_a(t), \theta_b(t))$ is from a small randomly selected subset of $\Delta$-fair set, $\forall t$ (to model the situation where perfect fairness is not feasible) and $\beta_a = \beta_b$. We observe that fairness is always violated at the beginning in lower plot even with small $\epsilon$. This is because the fairness set is found based on stable fixed points, which only concerns fairness in the long run.

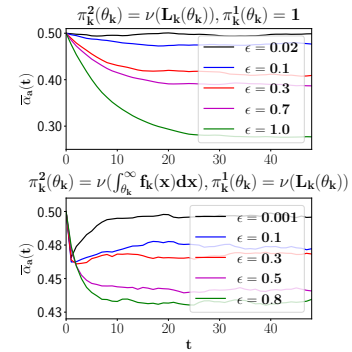

Fig. 7: Effect of $\Delta$-fair decisions found with proposed method.

We also trained binary classifiers over *Adult* dataset [4] by minimizing empirical loss where features are individual data points such as sex, race, and nationality, and labels are their annual income ($\geq 50k$ or $< 50k$). Since the dataset does not reflect dynamics, we employ (2) with $\pi_k(\theta_k) = \nu(L_k(\theta_k))$ and $\beta_a = \beta_b$. We examine the monotonic convergence of representation disparity under `Simple`, `EqOpt` (equalized false positive/negative cost(FPC/FNC)) and `EqLos`, and consider cases where $G_a$, $G_b$ are distinguished by the three features mentioned above. These results are shown in Fig. 8.

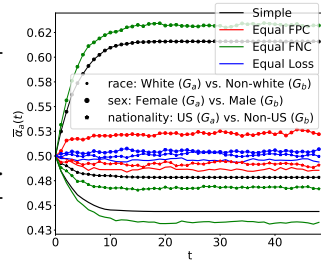

Fig. 8: Illustration of group representation disparity using *Adult* dataset.

## 5 Conclusion

This paper characterizes the impact of fairness intervention on group representation in a sequential setting. We show that the representation disparity can easily get exacerbated over time under relatively mild conditions. Our results suggest that fairness has to be defined with a good understanding of participation dynamics. Toward this end, we develop a method of selecting a proper fairness criterion based on prior knowledge of participation dynamics. Note that we do not always have full knowledge of participation dynamics; modeling dynamics from real-world measurements and finding a proper fairness criterion based on the obtained model is a potential direction for future work.

**Acknowledgments**

This work is supported by the NSF under grants CNS-1616575, CNS-1646019, CNS-1739517. The work of Cem Tekin was supported by BAGEP 2019 Award of the Science Academy.

## Footnotes

[2]This is a typical formulation if the objective $\boldsymbol{O}_t$ measures the average performance of decisions over all samples, i.e., $\boldsymbol{O}_t = \frac{1}{|G_a|+|G_b|}(\sum_{i \in G_a} O_t^i + \sum_{i \in G_b} O_t^i) = \frac{1}{|G_a|+|G_b|}(|G_a|O_{a,t} + |G_b|O_{b,t})$, where $O_t^i$ measures the performance of each sample $i$ and $O_{k,t} = \frac{1}{|G_k|}\sum_{i \in G_k} O_t^i$ is the average performance of $G_k$.

[3]This condition will always be satisfied when the system starts from a near empty state.

[4]Depending on the context, this criterion can also refer to equal false negative rate (FNR), true positive rate (TPR), or true negative rate (TNR), but the analysis is essentially the same.

[5] If the change of $f_{a,t}(x)$ and $f_{b,t}(x)$ w.r.t. the decisions follows the same rule (e.g., examples given in Section 3.3), then this relationship holds $\forall t$.

[6]The cases represented by blank cells cannot happen. When $\mathcal{C} = $ Simple, the table only illustrates the result when $\delta_{a,t}, \delta_{b,t} \in \mathcal{T}_{a,t} \cap \mathcal{T}_{b,t} \neq \emptyset$.

[7]When $f_{k,t}(x) = f_k(x)$, $\forall t$, subscript $t$ is omitted in some notations ($\phi_{\mathcal{C},t}, \delta_{k,t}, \pi_{k,t}$, etc.) for simplicity.

[8]By Fig. 3, we have $D_k(\theta) = g_k^1 - L_k(\theta)$.

[9]Here $L_{k,t}^1(\theta_k) = \int_{-\infty}^{\theta_k} f_{k,t}^1(x)dx$ and $L_{k,t}^0(\theta_k) = \int_{\theta_k}^{\infty} f_{k,t}^0(x)dx$.

[10]Suppose Assumption 1 holds for all $f_{k,t}^j(x)$ and their support does not change, then $f_{k,t}^1(x)$ and $f_{k,t}^0(x)$ overlap over $\mathcal{T}_k = [\underline{k}^1, \overline{k}^0]$, $\forall t$.

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
