[Supplementary Material]

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

# Appendix

## A  Notation table

| Notation | Description |
|---|---|
| $G_k, k \in \{a, b\}$ | two demographic groups |
| $G_k^j, j \in \{0, 1\}$ | subgroup with label $j$ in $G_k$ |
| $\alpha_k^j(t)$ | size of $G_k^j$ as a fraction of entire population at time $t$ |
| $\overline{\alpha}_k(t)$ | size of $G_k$ as a fraction of entire population at time $t$, i.e., $\alpha_a^0(t) + \alpha_k^1(t)$ |
| $g_{k,t}^j$ | fraction of subgroup with label $j$ in $G_k$ at time $t$, i.e., $\Pr(Y = j \mid K = k) = \alpha_k^j(t)/\overline{\alpha}_k(t)$ |
| $f_{k,t}^j(x)$ | feature distribution of $G_k^j$ at time $t$, i.e., $\Pr(X = x \mid K = k, Y = j)$ |
| $f_{k,t}(x)$ | feature distribution of $G_k$ at time $t$, i.e., $\Pr(X = x \mid K = k)$ and $f_{k,t}(x) = g_{k,t}^1 f_{k,t}^1(x) + g_{k,t}^0 f_{k,t}^0(x)$ |
| $h_\theta(x)$ | decision rule parameterized by $\theta$ |
| $\theta_k(t)$ | decision parameter for $G_k$ at time $t$ |
| $\boldsymbol{O}_t(\theta_a, \theta_b; \overline{\alpha}_a(t), \overline{\alpha}_b(t))$ | objective of one-shot problem at time $t$ with group proportions $\overline{\alpha}_a(t), \overline{\alpha}_b(t)$ |
| $O_{k,t}(\theta_k)$ | sub-objective of $G_k$ at time $t$ |
| $\Gamma_{\mathcal{C},t}(\theta_a, \theta_b)$ | a fairness constraint imposed on $\theta_a$ and $\theta_b$ for two groups at time $t$ |
| $N_k(t)$ | expected number of users from $G_k$ at time $t$ |
| $\pi_{k,t}(\theta_k(t))$ | retention rate of $G_k$ at time $t$ when imposing decision $\theta_k(t)$ |
| $\beta_k$ | number of exogenous arrivals to $G_k$ at every time step |
| $L_{k,t}(\theta_k)$ | expected loss incurred to $G_k$ by taking decision $\theta_k$ at time $t$ |
| $L_{k,t}^j(\theta_k)$ | expected loss incurred to $G_k^j$ by taking decision $\theta_k$ at time $t$ |
| $[\underline{k}_t^j, \overline{k}_t^j]$ | bounded support of distribution $f_{k,t}^j(x)$ |
| $\mathcal{T}_{k,t}$ | overlapping interval between $f_{k,t}^0(x)$ and $f_{k,t}^1(x)$ at time $t$, i.e., $[\underline{k}_t^1, \overline{k}_t^0]$ |
| $\delta_{k,t}$ | optimal decision for $G_k$ at time $t$ such that $\delta_{k,t} = \arg \min_\theta L_{k,t}(\theta)$ and satisfies $g_{k,t}^1 f_{k,t}^1(\delta_k) = g_{k,t}^0 f_{k,t}^0(\delta_k)$ |
| $\phi_{\mathcal{C},t}(\cdot)$ | a increasing function determined by constraint $\Gamma_{\mathcal{C},t}(\theta_a, \theta_b)$ mapping $\theta_b$ to $\theta_a$, i.e., $\Gamma_{\mathcal{C},t}(\phi_{\mathcal{C},t}(\theta_b), \theta_b)$ |
| $D_{k,t}(\theta_k)$ | intra-group disparity of $G_k$ at time $t$ |
| $\pi_k^m(\theta_k)$ | $m$th factor that drives user retention |
| $\Lambda(\cdot)$ | dynamics model specifying the relationship between $N_k(t+1)$ and $N_k(t+1), \beta_k, \pi_k^m(\theta_k(t))$ |

## B  Related Work

The impact of fairness interventions on both individuals and society, and the fairness in sequential decision making have been studied in the literature. [13] constructs a one-step feedback model over two consecutive time steps and characterizes the impact of fairness criteria (statistical parity and equal of opportunity) on changing each individual's feature and reshaping the entire population. Similarly, [9] proposes an effort-based measure of unfairness and constructs an individual-level model characterizing how an individual responds to the decisions based on it. The impact on the entire group is then derived from it and the impacts of fairness intervention are examined. While both highlight the importance of temporal modeling in evaluating the fairness, their main focus is on the adverse impact on feature distribution, rather than on group representation disparity. In contrast, our work focuses on the latter but also considers the impact of reshaping feature distributions. Moreover, we formulate the long-term impact over infinite horizon while [9, 13] only inspect the impact over two steps.

[7] also considers a sequential framework where the user departure is driven by model accuracy. It adopts the objective of minimizing the loss of the group with the highest loss (instead of overall or

average loss), which can prevent the extinction of any group from the system. It requires multiple demographic groups use the same model and does not adopt any fairness criterion. In contrast, we are more interested in the impact of various fairness criteria on representation disparity and if it is possible to sustain the group representation by imposing any fairness criterion. Other differences include the fact we consider the case when feature distributions are reshaped by the decisions (Section 3.3) and [7] does not.

[12] also constructs a two-stage model in the context of college admission, it shows that increasing admission rate of a group can increase the overall qualification for this group overtime. [10] describes a model in the context of labor market. They show that imposing the demographic parity constraint can incentivize under-represented groups to invest in education, which leads to a better long-term equilibrium.

Extensive studies on fairness in sequential decision making or online learning has been done [2, 3, 8, 11, 16, 17]. Most of them focus on proposing appropriate fairness notions to improve the fairness-accuracy trade-off. To the best of our knowledge, none of them considers the impact of fairness criteria on group representation disparity.

## C   Proof of Theorem 1

Theorem 1 is proved based on the following Lemma.

**Lemma 3.** *Let $a, b, z_a, z_b$ be real constants, where $a, b \in \mathbb{R}_+$ and $z_a, z_b \in [0, 1]$. If $b \geq a > 1$, $z_b - z_a > \frac{1}{a} - \frac{1}{b}$ and $b < \frac{1}{1-z_b}$ are satisfied, then the following holds:*

$$\frac{1 + z_a + az_a^2}{1 + z_b + bz_b^2} \leq \frac{1 + az_a}{1 + bz_b} \tag{5}$$

*Proof.* Re-organizing (5) gives the following:

$$(1 + z_a + az_a^2)(1 + bz_b) \leq (1 + z_b + bz_b^2)(1 + az_a)$$
$$bz_b + bz_a z_b + z_a + abz_a^2 z_b + az_a^2 \leq az_a + z_b + az_a z_b + bz_b^2 + abz_b^2 z_a$$

Proving (5) is equivalent to showing the following:

$$0 \leq (a-1)\frac{1}{z_b} + (1-b)\frac{1}{z_a} + b\frac{z_b}{z_a} - a\frac{z_a}{z_b} + \underbrace{a - b + ab(z_b - z_a)}_{\textbf{term 1}}$$

Since $z_b - z_a > \frac{1}{a} - \frac{1}{b}$, **term 1** $> a - b + b - a = 0$ holds. Therefore, proving (5) is equivalent to showing:

$$az_a^2 + (1-a)z_a \leq bz_b^2 + (1-b)z_b \tag{6}$$

Since $b < \frac{1}{1-z_b}$ holds, implying $z_b > 1 - \frac{1}{b}$.

Define a function $g(z) = cz^2 + (1-c)z$, $z \in [0, 1]$ under any constant $c > 1$. The following holds:

$$g(1 - \frac{1}{c}) = 0; \qquad g(1) = 1$$
$$g'(z) = 2cz + 1 - c; \quad g'(1 - \frac{1}{c}) = c - 1; \quad g''(z) = 2c$$

Since $g''(z)$ is a positive constant over $z \in [0, 1]$, $g'(z)$ is strictly increasing and $g'(z) > 0$ when $z \in (1 - \frac{1}{c}, 1]$, thus $g(z)$ is increasing over $z \in (1 - \frac{1}{c}, 1]$ from 0 to 1.

Now consider two functions $g_a(z) = az^2 + (1-a)z$ and $g_b(z) = bz^2 + (1-b)z$ with $z \in [0, 1]$. From the above analysis, $g_a(z)$ is increasing over $(1 - \frac{1}{a}, 1]$ from 0 to 1 and $g_b(z)$ is increasing over $(1 - \frac{1}{b}, 1]$ from 0 to 1. Moreover, $1 - \frac{1}{b} \geq 1 - \frac{1}{a}$ and $g_b''(z) = 2b \geq 2a = g_a''(z)$, i.e., the speed that $g_b(z)$ increases over $(1 - \frac{1}{b}, 1]$ is NOT slower than the speed that $g_a(z)$ increases over $(1 - \frac{1}{a}, 1]$. Since $z_b - z_a > \frac{1}{a} - \frac{1}{b} = (1 - \frac{1}{b}) - (1 - \frac{1}{a})$ and $z_b > 1 - \frac{1}{b}$, $g_a(z_a) \leq g_b(z_b)$ must hold.

Therefore, (6) is satisfied. Inequality (5) is proved.

$\square$

To simplify the notation, denote $\pi_{k,t} := \pi_{k,t}(\theta_k(t))$. We will only present the case when $\diamond :=$ " $<$ ", cases when $\diamond :=$ " $>$ " and $\diamond :=$ " $=$ " can be derived similarly and are omitted.

To prove Theorem 1, we prove the following statement using induction: If $\pi_{a,1} < \pi_{b,1}$, then $\forall t$, $\frac{\overline{\alpha}_a(t+1)}{\overline{\alpha}_b(t+1)} < \frac{\overline{\alpha}_a(t)}{\overline{\alpha}_b(t)}$ and $\pi_{a,t+1} < \pi_{a,t} < \pi_{b,t} < \pi_{b,t+1}$ hold under monotonicity condition. Moreover, $N_b(t) < \frac{\beta_b}{1-\pi_{b,t}}, \forall t$.

**Base Case:**

Since $\frac{N_a(1)}{N_b(1)} = \frac{\beta_a}{\beta_b}$. If $\pi_{a,1} < \pi_{b,1}$, then $\frac{\overline{\alpha}_a(2)}{\overline{\alpha}_b(2)} = \frac{N_a(1)\pi_{a,1}+\beta_a}{N_b(1)\pi_{b,1}+\beta_b} < \frac{N_a(1)}{N_b(1)} = \frac{\overline{\alpha}_a(1)}{\overline{\alpha}_b(1)}$. Under monotonicity condition, it results in $\pi_{a,2} < \pi_{a,1} < \pi_{b,1} < \pi_{b,2}$. Moreover, since $N_b(2) = N_b(1)\pi_{b,1} + \beta_b > N_b(1)$, implying $N_b(1) < \frac{\beta_b}{1-\pi_{b,1}}$.

**Induction Step:**

Suppose $\frac{\overline{\alpha}_a(t+1)}{\overline{\alpha}_b(t+1)} < \frac{\overline{\alpha}_a(t)}{\overline{\alpha}_b(t)} \leq \frac{\beta_a}{\beta_b}$, $\pi_{a,t+1} < \pi_{a,t} < \pi_{b,t} < \pi_{b,t+1}$ and $N_b(t) < \frac{\beta_b}{1-\pi_{b,t}}$ hold at time $t \geq 1$. Show that for time step $t+1$, $\frac{\overline{\alpha}_a(t+2)}{\overline{\alpha}_b(t+2)} < \frac{\overline{\alpha}_a(t+1)}{\overline{\alpha}_b(t+1)} \leq \frac{\beta_a}{\beta_b}$, $\pi_{a,t+2} < \pi_{a,t+1} < \pi_{b,t+1} < \pi_{b,t+2}$ and $N_b(t+1) < \frac{\beta_b}{1-\pi_{b,t+1}}$ also hold.

Denote $N_a(t) = c_a\beta_a$ and $N_b(t) = c_b\beta_b$. Since $N_k(t) = N_k(t-1)\pi_{k,t-1} + \beta_k > \beta_k, \forall t$, it holds that $c_a, c_b > 1$.

By hypothesis, $\frac{\overline{\alpha}_a(t)}{\overline{\alpha}_b(t)} \leq \frac{\beta_a}{\beta_b}$ implies that $c_b \geq c_a > 1$, and $N_b(t) < \frac{\beta_b}{1-\pi_{b,t}}$ implies that $c_b < \frac{1}{1-\pi_{b,t}}$. Since $\frac{N_a(t+1)}{N_b(t+1)} = \frac{N_a(t)\pi_{a,t}+\beta_a}{N_b(t)\pi_{b,t}+\beta_b} = \frac{\beta_a}{\beta_b}\frac{c_a\pi_{a,t}+1}{c_b\pi_{b,t}+1} < \frac{N_a(t)}{N_b(t)} = \frac{\beta_a}{\beta_b}\frac{c_a}{c_b}$, re-organizing it gives $\pi_{b,t} - \pi_{a,t} > \frac{1}{c_a} - \frac{1}{c_b}$.

By Lemma 3, the following holds:

$$\frac{N_a(t)\pi_{a,t}^2 + \beta_a(1+\pi_{a,t})}{N_b(t)\pi_{b,t}^2 + \beta_b(1+\pi_{b,t})} = \frac{\beta_a}{\beta_b}\frac{1+\pi_{a,t}+c_a\pi_{a,t}^2}{1+\pi_{b,t}+c_b\pi_{b,t}^2} \leq \frac{\beta_a}{\beta_b}\frac{1+c_a\pi_{a,t}}{1+c_b\pi_{b,t}} = \frac{N_a(t+1)}{N_b(t+1)} = \frac{\overline{\alpha}_a(t+1)}{\overline{\alpha}_b(t+1)}$$

Since we suppose $\pi_{a,t+1} < \pi_{a,t} < \pi_{b,t} < \pi_{b,t+1}$, we have:

$$\frac{N_a(t)\pi_{a,t}^2 + \beta_a(1+\pi_{a,t})}{N_b(t)\pi_{b,t}^2 + \beta_b(1+\pi_{b,t})} > \frac{(N_a(t)\pi_{a,t}+\beta_a)\pi_{a,t+1}+\beta_a}{(N_b(t)\pi_{b,t}+\beta_b)\pi_{b,t+1}+\beta_b} = \frac{\overline{\alpha}_a(t+2)}{\overline{\alpha}_b(t+2)}$$

It implies that $\frac{\overline{\alpha}_a(t+2)}{\overline{\alpha}_b(t+2)} < \frac{\overline{\alpha}_a(t+1)}{\overline{\alpha}_b(t+1)}$.

By motonoticity condition, it results in $\pi_{a,t+2} < \pi_{a,t+1} < \pi_{b,t+1} < \pi_{b,t+2}$.

Moreover, $N_b(t+1) = N_b(t)\pi_{b,t} + \beta_b < \frac{\beta_b\pi_{b,t}}{1-\pi_{b,t}} + \beta_b = \frac{\beta_b}{1-\pi_{b,t}} < \frac{\beta_b}{1-\pi_{b,t+1}}$.

The statement holds for time $t+1$. This completes the proof.

# D Proof of Theorem 2

Without loss of generality, let $\frac{\widehat{\overline{\alpha}}_a}{\widehat{\overline{\alpha}}_b} < \frac{\widetilde{\overline{\alpha}}_a}{\widetilde{\overline{\alpha}}_b}$. Since $\pi_k(\theta_k) = h_k(O_k(\theta_k))$ with $h_k(\cdot)$ being a decreasing function, showing that $\widetilde{\mathbf{O}}$ and $\widehat{\mathbf{O}}$ satisfy Monotonicity condition is equivalent to showing that $O_a(\widehat{\theta}_a) > O_a(\widetilde{\theta}_a)$, $O_b(\widehat{\theta}_b) < O_b(\widetilde{\theta}_b)$. Under the condition that $O_k(\widehat{\theta}_k) \neq O_k(\widetilde{\theta}_k)$ for any possible $\widehat{\overline{\alpha}}_a \neq \widetilde{\overline{\alpha}}_a$, prove by contradiction: suppose $O_a(\widehat{\theta}_a) < O_a(\widetilde{\theta}_a)$ holds, then $O_b(\widehat{\theta}_b) > O_b(\widetilde{\theta}_b)$ must also hold otherwise $(\widehat{\theta}_a, \widehat{\theta}_b)$ will be the solution to $\widetilde{\mathbf{O}}$.

Because $(\widehat{\theta}_a, \widehat{\theta}_b)$ is the optimal solution to $\widehat{\mathbf{O}}$ and $(\widetilde{\theta}_a, \widetilde{\theta}_b)$ is the optimal solution to $\widetilde{\mathbf{O}}$, and $O_b(\widehat{\theta}_b) > O_b(\widetilde{\theta}_b)$, the following holds:

$$\widehat{\overline{\alpha}}_a O_a(\widehat{\theta}_a) + \widehat{\overline{\alpha}}_b O_b(\widehat{\theta}_b) \le \widehat{\overline{\alpha}}_a O_a(\widetilde{\theta}_a) + \widehat{\overline{\alpha}}_b O_b(\widetilde{\theta}_b) \to \frac{O_a(\widehat{\theta}_a) - O_a(\widetilde{\theta}_a)}{O_b(\widetilde{\theta}_b) - O_b(\widehat{\theta}_b)} \ge \frac{\widehat{\overline{\alpha}}_b}{\widehat{\overline{\alpha}}_a}$$

$$\widetilde{\overline{\alpha}}_a O_a(\widetilde{\theta}_a) + \widetilde{\overline{\alpha}}_b O_b(\widetilde{\theta}_b) \le \widetilde{\overline{\alpha}}_a O_a(\widehat{\theta}_a) + \widetilde{\overline{\alpha}}_b O_b(\widehat{\theta}_b) \to \frac{O_a(\widehat{\theta}_a) - O_a(\widetilde{\theta}_a)}{O_b(\widetilde{\theta}_b) - O_b(\widehat{\theta}_b)} \le \frac{\widetilde{\overline{\alpha}}_b}{\widetilde{\overline{\alpha}}_a}$$

It implies that $\frac{\widehat{\overline{\alpha}}_a}{\widehat{\overline{\alpha}}_b} \ge \frac{\widetilde{\overline{\alpha}}_a}{\widetilde{\overline{\alpha}}_b}$, which is a contradiction.

# E   Proof of Lemma 1

Starting from Appendix E until Appendix H, we simplify the notations by removing $t$ from subscript, i.e., $L_{k,t}(\theta_k) := L_k(\theta_k)$, $g_{k,t}^j := g_k^j$, $f_{k,t}(x) := f_k(x)$, $f_{k,t}^j(x) := f_k^j(x)$, $\underline{k}_t^j := \underline{k}^j$, $\overline{k}_t^j := \overline{k}^j$, $\phi_{\mathcal{C},t} := \phi_{\mathcal{C}}$, $\Gamma_{\mathcal{C},t} := \Gamma_{\mathcal{C}}$, $\delta_{k,t} := \delta_k$, $\mathcal{T}_{k,t} := \mathcal{T}_k$.

The loss for group $k$ can be written as

$$L_k(\theta_k) = \int_{-\infty}^{\theta_k} g_k^1 f_k^1(x)dx + \int_{\theta_k}^{\infty} g_k^0 f_k^0(x)dx = \begin{cases} \int_{\theta_k}^{\overline{k}^0} g_k^0 f_k^0(x)dx, \text{ if } \theta_k \in [\underline{k}^0, \underline{k}^1] \\ \int_{\theta_k}^{\overline{k}^0} g_k^0 f_k^0(x)dx + \int_{\underline{k}^1}^{\theta_k} g_k^1 f_k^1(x)dx, \text{ if } \theta_k \in [\underline{k}^1, \overline{k}^0] \\ \int_{\underline{k}^1}^{\theta_k} g_k^1 f_k^1(x)dx, \text{ if } \theta_k \in [\overline{k}^0, \overline{k}^1] \end{cases}$$

which is decreasing in $\theta_k$ over $[\underline{k}^0, \underline{k}^1]$ and increasing over $[\overline{k}^0, \overline{k}^1]$, the optimal solution $\theta_k^* \in [\underline{k}^1, \overline{k}^0]$. Taking derivative of $L_k(\theta_k)$ w.r.t. $\theta_k$ gives $\frac{dL_k(\theta_k)}{d\theta_k} = g_k^1 f_k^1(\theta_k) - g_k^0 f_k^0(\theta_k)$, which is strictly increasing over $[\underline{k}^1, \overline{k}^0]$ under Assumption 1.

The optimal solution $\theta_k^* = \arg\min_{\theta_k} L_k(\theta_k) \in \{\underline{k}^1, \delta_k, \overline{k}^0\}$ can be thus found easily. Moreover, $L_k(\theta_k)$ is decreasing in $\theta_k$ over $[\underline{k}^0, \theta_k^*]$ and increasing over $[\theta_k^*, \overline{k}^1]$.

# F   Proof of Lemma 2

Some notations are simplified by removing subscript $t$ as mentioned in Appendix E.

We proof this Lemma by contradiction.

Let $\mathcal{V} = \{(\theta_a, \theta_b) | \theta_a \in [\phi_{\mathcal{C}}(\delta_b), \delta_a], \theta_b \in [\delta_b, \phi_{\mathcal{C}}^{-1}(\delta_a)], \Gamma_{\mathcal{C}}(\theta_a, \theta_b) = 0\}$.

Note that for `Simple`, `EqOpt`, `StatPar` fairness, for any $(\theta_a, \theta_b)$ and $(\theta_a', \theta_b')$ that satisfy constraints $\Gamma_{\mathcal{C}}(\theta_a, \theta_b) = 0$ and $\Gamma_{\mathcal{C}}(\theta_a', \theta_b') = 0$, $\theta_a \ge \theta_a'$ if and only if $\theta_b \ge \theta_b'$. Suppose that $(\check{\theta}_a, \check{\theta}_b)$ satisfies $\Gamma_{\mathcal{C}}(\check{\theta}_a, \check{\theta}_b) = 0$ and $(\check{\theta}_a, \check{\theta}_b) = \arg\min_{\theta_a, \theta_b} \overline{\alpha}_a L_a(\theta_a) + \overline{\alpha}_b L_b(\theta_b) \notin \mathcal{V}$, then one of the following must hold: (1) $\check{\theta}_a < \phi_{\mathcal{C}}(\delta_b)$, $\check{\theta}_b < \delta_b$; (2) $\check{\theta}_a > \delta_a$, $\check{\theta}_b > \phi_{\mathcal{C}}^{-1}(\delta_a)$. Consider two cases separately.

(1) $\check{\theta}_a < \phi_{\mathcal{C}}(\delta_b)$, $\check{\theta}_b < \delta_b$

Since $L_b(\check{\theta}_b) > L_b(\delta_b)$, $\forall \overline{\alpha}_a, \overline{\alpha}_b$, to satisfy $\overline{\alpha}_a L_a(\check{\theta}_a) + \overline{\alpha}_b L_b(\check{\theta}_b) < \overline{\alpha}_a L_a(\phi(\delta_b)) + \overline{\alpha}_b L_b(\delta_b)$, $L_a(\check{\theta}_a) < L_a(\phi_{\mathcal{C}}(\delta_b))$ must hold. However, by Lemma 1, $L_a(\theta_a)$ is strictly decreasing on $[\underline{a}^0, \delta_a]$ and strictly increasing on $[\delta_a, \overline{a}^1]$. Since $\check{\theta}_a < \phi_{\mathcal{C}}(\delta_b) < \delta_a$, this implies $L_a(\check{\theta}_a) > L_a(\phi_{\mathcal{C}}(\delta_b))$. Therefore, $(\check{\theta}_a, \check{\theta}_b)$ cannot be the optimal pair.

(2) $\check{\theta}_a > \delta_a$, $\check{\theta}_b > \phi_{\mathcal{C}}^{-1}(\delta_a)$

Since $L_a(\check{\theta}_a) > L_a(\delta_a)$, $\forall \overline{\alpha}_a, \overline{\alpha}_b$, to satisfy $\overline{\alpha}_a L_a(\check{\theta}_a) + \overline{\alpha}_b L_b(\check{\theta}_b) < \overline{\alpha}_a L_a(\delta_a) + \overline{\alpha}_b L_b(\phi_{\mathcal{C}}^{-1}(\delta_a))$, $L_b(\check{\theta}_b) < L_b(\phi_{\mathcal{C}}^{-1}(\delta_a))$ must hold. However, by Lemma 1, $L_b(\theta_b)$ is strictly decreasing on $[\underline{b}^0, \delta_b]$ and strictly increasing on $[\delta_b, \overline{b}^1]$. Since $\check{\theta}_b > \phi_{\mathcal{C}}^{-1}(\delta_a) > \delta_b$, this implies $L_b(\check{\theta}_b) > L_b(\phi_{\mathcal{C}}^{-1}(\delta_a))$. Therefore, $(\check{\theta}_a, \check{\theta}_b)$ cannot be the optimal pair.

# G Proof of Theorem 3

Some notations are simplified by removing subscript $t$ as mentioned in Appendix E.

Proof of Theorem 3 is based on the following Lemma.

**Lemma 4.** *Consider the one-shot problem* (1) *at some time step $t$, with group proportions given by $\overline{\alpha}_a(t), \overline{\alpha}_b(t)$. Under Assumption 1 the one-shot decision $(\theta_a(t), \theta_b(t))$ for this time step is unique and satisfies the following:*

*(1) Under* `EqOpt` *fairness:*

- *If $\theta_a(t) \in [\underline{a}^0, \underline{a}^1]$, $\theta_b(t) \in [\underline{b}^1, \overline{b}^0]$, then $\frac{\overline{\alpha}_a(t)}{\overline{\alpha}_b(t)} = \left( \frac{g_b^1}{g_b^0} \frac{f_b^1(\theta_b(t))}{f_b^0(\theta_b(t))} - 1 \right) \frac{g_b^0}{g_a^0}$.*

- *If $\theta_a(t) \in [\underline{a}^1, \overline{a}^0]$, $\theta_b(t) \in [\underline{b}^1, \overline{b}^0]$, then $\frac{\overline{\alpha}_a(t)}{\overline{\alpha}_b(t)} = \frac{\frac{g_b^1}{g_b^0} \frac{f_b^1(\theta_b(t))}{f_b^0(\theta_b(t))} - 1}{1 - \frac{g_a^1}{g_a^0} \frac{f_a^1(\theta_a(t))}{f_a^0(\theta_a(t))}} \frac{g_b^0}{g_a^0}$.*

*(2) Under* `StatPar` *fairness:*

- *If $\theta_a(t) \in [\underline{a}^0, \underline{a}^1]$, $\theta_b(t) \in [\underline{b}^1, \overline{b}^0]$, then $\frac{\overline{\alpha}_a(t)}{\overline{\alpha}_b(t)} = 1 - \frac{2}{\frac{g_b^1}{g_b^0} \frac{f_b^1(\theta_b(t))}{f_b^0(\theta_b(t))} + 1}$.*

- *If $\theta_a(t) \in [\underline{a}^1, \overline{a}^0]$, $\theta_b(t) \in [\underline{b}^1, \overline{b}^0]$, then $\frac{\overline{\alpha}_a(t)}{\overline{\alpha}_b(t)} = \left( 1 - \frac{2}{\frac{g_b^1}{g_b^0} \frac{f_b^1(\theta_b(t))}{f_b^0(\theta_b(t))} + 1} \right) \left( \frac{2}{1 - \frac{g_a^1 f_a^1(\theta_a(t))}{g_a^0 f_a^0(\theta_a(t))}} - 1 \right)$.*

- *If $\theta_a(t) \in [\underline{a}^1, \overline{a}^0]$, $\theta_b(t) \in [\overline{b}^0, \overline{b}^1]$, then $\frac{\overline{\alpha}_a(t)}{\overline{\alpha}_b(t)} = \frac{2}{1 - \frac{g_a^1 f_a^1(\theta_a(t))}{g_a^0 f_a^0(\theta_a(t))}} - 1$.*

*(3) Under* `Simple` *fairness:*

- *If we further assume $\delta_a, \delta_b \in \mathcal{T}_a \cap \mathcal{T}_b$, then $\theta_a(t) = \theta_b(t) \in [\underline{a}^1, \overline{b}^0]$ and $\frac{\overline{\alpha}_a(t)}{\overline{\alpha}_b(t)} = \frac{g_b^1 f_b^1(\theta_b(t)) - g_b^0 f_b^0(\theta_b(t))}{g_a^0 f_a^0(\theta_a(t)) - g_a^1 f_a^1(\theta_a(t))}$.*

*Proof.* We focus on the case when $g_b^1 f_a^1(\underline{a}^1) < g_a^0 f_a^0(\underline{a}^1)$ & $g_b^1 f_a^1(\overline{a}^0) > g_a^0 f_a^0(\overline{a}^0)$ and $g_b^1 f_b^1(\underline{b}^1) < g_b^0 f_b^0(\underline{b}^1)$ & $g_b^1 f_b^1(\overline{b}^0) > g_b^0 f_b^0(\overline{b}^0)$. That is, $\theta_k^* = \arg\min_\theta L_k(\theta) = \delta_k$ holds for $k \in \{a, b\}$.

Constraint $\Gamma_{\mathcal{C}}(\theta_a, \theta_b) = 0$ can be rewritten as $\theta_a = \phi_{\mathcal{C}}(\theta_b)$ for some strictly increasing function $\phi_{\mathcal{C}}$. The following holds:

$$\frac{d\phi_{\mathcal{C}}(\theta_b)}{d\theta_b} = -\frac{\frac{\partial \Gamma_{\mathcal{C}}(\theta_a, \theta_b)}{\partial \theta_b}}{\frac{\partial \Gamma_{\mathcal{C}}(\theta_a, \theta_b)}{\partial \theta_a}} \Big|_{\theta_a = \phi_{\mathcal{C}}(\theta_b)} = \begin{cases} \frac{f_b^0(\theta_b)}{f_a^0(\phi_{\mathcal{C}}(\theta_b))}, & \mathcal{C} := \text{EqOpt} \\ \frac{g_b^0 f_b^0(\theta_b) + g_b^1 f_b^1(\theta_b)}{g_a^0 f_a^0(\phi_{\mathcal{C}}(\theta_b)) + g_a^1 f_a^1(\phi_{\mathcal{C}}(\theta_b))}, & \mathcal{C} := \text{StaPar} \\ 1, & \mathcal{C} := \text{Simple} \end{cases}$$

The one-shot problem can be expressed with only one variable, either $\theta_a$ or $\theta_b$. Here we express it in terms of $\theta_b$. At each round, decision maker finds $\theta_b(t) = \arg\min_{\theta_b} L^t(\theta_b) = \overline{\alpha}_a(t) L_a(\phi_{\mathcal{C}}(\theta_b)) + \overline{\alpha}_b(t) L_b(\theta_b)$ and $\theta_a(t) = \phi_{\mathcal{C}}(\theta_b(t))$. Since $\phi_{\mathcal{C}}(\delta_b) < \delta_a$ ( $\phi_{\mathcal{C}}^{-1}(\delta_a) > \delta_b$), when $\mathcal{C} := \text{StatPar}$, solution $(\theta_a(t), \theta_b(t))$ can be in one of the following three forms: (1) $\theta_a(t) \in [\underline{a}^0, \underline{a}^1]$, $\theta_b(t) \in [\underline{b}^1, \overline{b}^0]$; (2) $\theta_a(t) \in [\underline{a}^1, \overline{a}^0]$, $\theta_b(t) \in [\underline{b}^1, \overline{b}^0]$; (3) $\theta_a(t) \in [\underline{a}^1, \overline{a}^0]$, $\theta_b(t) \in [\overline{b}^0, \overline{b}^1]$. When $\mathcal{C} := \text{EqOpt}$, solution $(\theta_a(t), \theta_b(t))$ can be either (1) or (2) listed above. In the following analysis, we simplify the notation $\phi_{\mathcal{C}}$ as $\phi$ when fairness criterion $\mathcal{C}$ is explicitly stated. For `EqOpt` and `StatPar` criteria, we consider each case separately.

**Case 1:** $\theta_a(t) \in [\underline{a}^0, \underline{a}^1]$, $\theta_b(t) \in [\underline{b}^1, \overline{b}^0]$

Let $\theta_b^{\max} = \min\{\overline{b}^0, \phi_{\mathcal{C}}^{-1}(\underline{a}^1)\}$ be the maximum value $\theta_b$ can take. $L^t(\theta_b) = \overline{\alpha}_b(t) \int_{\underline{b}^1}^{\theta_b} g_b^1 f_b^1(x) - g_b^0 f_b^0(x) dx - \overline{\alpha}_a(t) \int_{\underline{a}^0}^{\phi_{\mathcal{C}}(\theta_b)} g_a^0 f_a^0(x) dx + \overline{\alpha}_a(t) g_a^0 + \overline{\alpha}_b(t) \int_{\underline{b}^1}^{\overline{b}^0} g_b^0 f_b^0(x) dx$

Taking derivative w.r.t. $\theta_b$ gives

$$\frac{dL^t(\theta_b)}{d\theta_b} = \overline{\alpha}_b(t) (g_b^1 f_b^1(\theta_b) - g_b^0 f_b^0(\theta_b)) - \overline{\alpha}_a(t) g_a^0 f_a^0(\phi_{\mathcal{C}}(\theta_b)) \frac{d\phi_{\mathcal{C}}(\theta_b)}{d\theta_b}.$$

1. $\mathcal{C} := \texttt{EqOpt}$

$\frac{dL^t(\theta_b)}{d\theta_b} = \overline{\alpha}_b(t)(g_b^1 f_b^1(\theta_b) - g_b^0 f_b^0(\theta_b)) - \overline{\alpha}_a(t)g_a^0 f_b^0(\theta_b)$, since $g_b^1 f_b^1(\theta_b) - g_b^0 f_b^0(\theta_b)$ is increasing from negative to positive and $f_b^0(\theta_b)$ is decreasing over $[\underline{b}^1, \overline{b}^0]$, implying $\frac{dL^t(\theta_b)}{d\theta_b}$ is increasing over $[\underline{b}^1, \overline{b}^0]$. Based on the value of $\frac{\overline{\alpha}_a(t)}{\overline{\alpha}_b(t)}$,

- If $\frac{dL^t(\theta_b)}{d\theta_b}|_{\theta_b = \theta_b^{\max}} \geq 0$, then one-shot decision $\theta_b(t)$ satisfies $\frac{\overline{\alpha}_a(t)}{\overline{\alpha}_b(t)} = (\frac{g_b^1}{g_b^0} \frac{f_b^1(\theta_b(t))}{f_b^0(\theta_b(t))} - 1)\frac{g_b^0}{g_a^0}$ and is unique.

- If $\frac{dL^t(\theta_b)}{d\theta_b} < 0, \forall \theta_b \in [\underline{b}^1, \theta_b^{\max}]$, then $\theta_b(t) > \theta_b^{\max}$ and $(\theta_a(t), \theta_b(t))$ does not satisfy Case 1.

2. $\mathcal{C} := \texttt{StatPar}$

$\frac{dL^t(\theta_b)}{d\theta_b} = \overline{\alpha}_b(t)(g_b^1 f_b^1(\theta_b) - g_b^0 f_b^0(\theta_b)) - \overline{\alpha}_a(t)\frac{g_b^1 f_b^1(\theta_b) + g_b^0 f_b^0(\theta_b)}{1 + \frac{g_a^1 f_a^1(\phi(\theta_b))}{g_a^0 f_a^0(\phi(\theta_b))}} = (\overline{\alpha}_b(t) - \overline{\alpha}_a(t))g_b^1 f_b^1(\theta_b) -$ $(\overline{\alpha}_b(t) + \overline{\alpha}_a(t))g_b^0 f_b^0(\theta_b)$, where the last equality holds since $f_a^1(\phi(\theta_b)) = 0$ over $[\underline{a}^0, \underline{a}^1]$. Since $\frac{dL^t(\theta_b)}{d\theta_b}|_{\theta_b = \underline{b}^1} < 0$, based on the value of $\frac{\overline{\alpha}_a(t)}{\overline{\alpha}_b(t)}$,

- If $\exists \theta_b'$ such that $\frac{dL^t(\theta_b)}{d\theta_b}|_{\theta_b = \theta_b'} \geq 0$, then one-shot decision $\theta_b(t)$ satisfies $\frac{\overline{\alpha}_a(t)}{\overline{\alpha}_b(t)} = 1 - \frac{2}{\frac{g_b^1}{g_b^0}\frac{f_b^1(\theta_b(t))}{f_b^0(\theta_b(t))} + 1}$

and is unique.

- If $\frac{dL^t(\theta_b)}{d\theta_b} < 0, \forall \theta_b \in [\underline{b}^1, \theta_b^{\max}]$, then $\theta_b(t) > \theta_b^{\max}$ and $(\theta_a(t), \theta_b(t))$ does not satisfy Case 1.

**Case 2:** $\theta_a(t) \in [\underline{a}^1, \overline{a}^0], \theta_b(t) \in [\underline{b}^1, \overline{b}^0]$

Let $\theta_b^{\max} = \min\{\overline{b}^0, \phi_{\mathcal{C}}^{-1}(\overline{a}^0)\}$ and $\theta_b^{\min} = \max\{\underline{b}^1, \phi_{\mathcal{C}}^{-1}(\underline{a}^1)\}$ be the maximum and minimum value that $\theta_b$ can take respectively. $L^t(\theta_b) = \overline{\alpha}_b(t)\int_{\underline{b}^1}^{\theta_b} g_b^1 f_b^1(x) - g_b^0 f_b^0(x)dx + \overline{\alpha}_a(t)\int_{\underline{a}^1}^{\phi_{\mathcal{C}}(\theta_b)} g_a^1 f_a^1(x) -$ $g_a^0 f_a^0(x)dx + \overline{\alpha}_b(t)\int_{\underline{b}^1}^{\overline{b}^0} g_b^0 f_b^0(x)dx + \overline{\alpha}_a(t)\int_{\underline{a}^1}^{\overline{a}^0} g_a^0 f_a^0(x)dx$

Taking derivative w.r.t. $\theta_b$ gives

$$\frac{dL^t(\theta_b)}{d\theta_b} = \overline{\alpha}_b(t)(g_b^1 f_b^1(\theta_b) - g_b^0 f_b^0(\theta_b)) + \overline{\alpha}_a(t)(g_a^1 f_a^1(\phi_{\mathcal{C}}(\theta_b)) - g_a^0 f_a^0(\phi_{\mathcal{C}}(\theta_b)))\frac{d\phi_{\mathcal{C}}(\theta_b)}{d\theta_b}.$$

1. $\mathcal{C} := \texttt{EqOpt}$

$\frac{dL^t(\theta_b)}{d\theta_b} = ((g_a^1 \frac{f_a^1(\phi(\theta_b))}{f_a^0(\phi(\theta_b))} - g_a^0)\overline{\alpha}_a(t) - g_b^0\overline{\alpha}_b(t))f_b^0(\theta_b) + g_b^1 f_b^1(\theta_b)\overline{\alpha}_b(t)$. Since $\frac{dL^t(\theta_b)}{d\theta_b}|_{\theta_b = \theta_b^{\max}} > 0$, based on $\frac{\overline{\alpha}_a(t)}{\overline{\alpha}_b(t)}$,

- If $\exists \theta_b'$ such that $\frac{dL^t(\theta_b)}{d\theta_b}|_{\theta_b = \theta_b'} \leq 0$, then one-shot decision $\theta_b(t)$ satisfies $\frac{\overline{\alpha}_a(t)}{\overline{\alpha}_b(t)} = \frac{1 - \frac{g_b^1}{g_b^0}\frac{f_b^1(\theta_b(t))}{f_b^0(\theta_b(t))}}{\frac{g_a^1}{g_a^0}\frac{f_a^1(\phi(\theta_b(t)))}{f_a^0(\phi(\theta_b(t)))} - 1}\frac{g_b^0}{g_a^0}$

and is unique.

- If $\frac{dL^t(\theta_b)}{d\theta_b} > 0, \forall \theta_b \in [\theta_b^{\min}, \theta_b^{\max}]$, then $\theta_b(t) < \theta_b^{\min}$ and $(\theta_a(t), \theta_b(t))$ does not satisfy Case 2.

2. $\mathcal{C} := \texttt{StatPar}$

$\frac{dL^t(\theta_b)}{d\theta_b} = \overline{\alpha}_b(t)(g_b^1 f_b^1(\theta_b) - g_b^0 f_b^0(\theta_b)) + \overline{\alpha}_a(t)(g_b^0 f_b^0(\theta_b) + g_b^1 f_b^1(\theta_b))\frac{g_a^1 f_a^1(\phi(\theta_b)) - g_a^0 f_a^0(\phi(\theta_b))}{g_a^1 f_a^1(\phi(\theta_b)) + g_a^0 f_a^0(\phi(\theta_b))}$.

- If $\exists \theta_b(t)$ such that $\frac{dL^t(\theta_b)}{d\theta_b}|_{\theta_b = \theta_b(t)} = 0$, then it satisfies $\frac{\overline{\alpha}_a(t)}{\overline{\alpha}_b(t)} = (1 - \frac{2}{\frac{g_b^1}{g_b^0}\frac{f_b^1(\theta_b(t))}{f_b^0(\theta_b(t))} + 1})(\frac{2}{1 - \frac{g_a^1 f_a^1(\phi(\theta_b(t)))}{g_a^0 f_a^0(\phi(\theta_b(t)))}} - 1)$ and is unique.

- If $\frac{dL^t(\theta_b)}{d\theta_b} > 0, \forall \theta_b \in [\theta_b^{\min}, \theta_b^{\max}]$, then $\theta_b(t) < \theta_b^{\min}$ and $(\theta_a(t), \theta_b(t))$ does not satisfy Case 2.

- If $\frac{dL^t(\theta_b)}{d\theta_b} < 0, \forall \theta_b \in [\theta_b^{\min}, \theta_b^{\max}]$, then $\theta_b(t) > \theta_b^{\max}$ and $(\theta_a(t), \theta_b(t))$ does not satisfy Case 2.

**Case 3:** $\theta_a(t) \in [\underline{a}^1, \overline{a}^0], \theta_b(t) \in [\overline{b}^0, \overline{b}^1]$

Express $L^t(\theta_a, \theta_b)$ as function of $\theta_a$, the analysis will be similar to Case 1.

Let $\theta_a^{\min} = \max\{\underline{a}^1, \phi_{\mathcal{C}}(\overline{b}^0)\}$ be the minimum value $\theta_a$ can take.

$L^t(\theta_a) = \overline{\alpha}_a(t) \int_{\underline{a}^1}^{\theta_a} g_a^1 f_a^1(x) - g_a^0 f_a^0(x) dx + \overline{\alpha}_b(t) \int_{\underline{b}^1}^{\phi_{\mathcal{C}}^{-1}(\theta_a)} g_b^1 f_b^1(x) dx + \overline{\alpha}_a(t) \int_{\underline{a}^1}^{\overline{a}^0} g_a^0 f_a^0(x) dx$

Taking derivative w.r.t. $\theta_a$ gives

$$\frac{dL^t(\theta_a)}{d\theta_a} = \overline{\alpha}_a(t)(g_a^1 f_a^1(\theta_a) - g_a^0 f_a^0(\theta_a)) + \overline{\alpha}_b(t) g_b^1 f_b^1(\phi_{\mathcal{C}}^{-1}(\theta_a)) \frac{d\phi_{\mathcal{C}}^{-1}(\theta_a)}{d\theta_a},$$

where $\mathcal{C} :=$ `StatPar`.

$\frac{dL^t(\theta_a)}{d\theta_a} = \overline{\alpha}_a(t)(g_a^1 f_a^1(\theta_a) - g_a^0 f_a^0(\theta_a)) + \overline{\alpha}_b(t) \frac{g_a^1 f_a^1(\theta_a) + g_a^0 f_a^0(\theta_a)}{1 + \frac{g_b^0 f_b^0(\phi^{-1}(\theta_a))}{g_b^0 f_b^0(\phi^{-1}(\theta_a))}} = \overline{\alpha}_a(t)(g_a^1 f_a^1(\theta_a) - g_a^0 f_a^0(\theta_a)) +$

$\overline{\alpha}_b(t)(g_a^1 f_a^1(\theta_a) + g_a^0 f_a^0(\theta_a))$, where the last equality holds since $f_b^0(\phi^{-1}(\theta_a)) = 0$ over $[\overline{b}^0, \overline{b}^1]$. Since $\frac{dL^t(\theta_b)}{d\theta_b}|_{\theta_b = \overline{a}^0} > 0$, based on the value of $\frac{\overline{\alpha}_a(t)}{\overline{\alpha}_b(t)}$,

- If $\exists \theta'_a$ such that $\frac{dL^t(\theta_a)}{d\theta_a}|_{\theta_a = \theta'_a} \leq 0$, then one-shot decision $\theta_a(t)$ satisfies $\frac{\overline{\alpha}_a(t)}{\overline{\alpha}_b(t)} = \frac{2}{1 - \frac{g_a^1 f_a^1(\theta_a(t))}{g_a^0 f_a^0(\theta_a(t))}} - 1$

and is unique.

- If $\frac{dL^t(\theta_a)}{d\theta_a} > 0, \forall \theta_a \in [\underline{b}^1, \theta_a^{\min}]$, then $\theta_a(t) < \theta_a^{\min}$ and $(\theta_a(t), \theta_b(t))$ does not satisfy Case 3.

Now consider the case when $\mathcal{C} :=$ `Simple`, where $\theta_a(t) = \theta_b(t) = \theta(t)$. Since $\delta_a > \delta_b$, suppose that both $\delta_a, \delta_b \in \mathcal{T}_a \cap \mathcal{T}_b$ and according to Lemma 2, there could be only one case: $\theta(t) \in [\underline{a}^1, \overline{b}^0]$.

Taking derivative w.r.t. $\theta$ gives

$$\frac{dL^t(\theta)}{d\theta} = \overline{\alpha}_b(t)(g_b^1 f_b^1(\theta) - g_b^0 f_b^0(\theta)) + \overline{\alpha}_a(t)(g_a^1 f_a^1(\theta) - g_a^0 f_a^0(\theta)).$$

$\frac{dL^t(\theta)}{d\theta}$ is increasing from negative to positive over $[\delta_b, \delta_a]$, $\exists \theta(t)$ such that $\frac{dL^t(\theta)}{d\theta}|_{\theta = \theta(t)} = 0$, and it satisfies $\frac{\overline{\alpha}_a(t)}{\overline{\alpha}_b(t)} = \frac{g_b^1 f_b^1(\theta(t)) - g_b^0 f_b^0(\theta(t))}{g_a^0 f_a^0(\theta(t)) - g_a^1 f_a^1(\theta(t))}$.

$\square$

By Lemma 2, $\theta_a(t) \in [\phi_{\mathcal{C}}(\delta_b), \delta_a], \theta_b(t) \in [\delta_b, \phi_{\mathcal{C}}^{-1}(\delta_a)]$ hold. Under Assumption 1, $f_b^1 f_b^1(\theta_b) \geq f_b^0 f_b^0(\theta_b)$ for $\theta_b \in [\delta_b, \overline{b}^0]$, $f_a^1 f_a^1(\theta_a) \leq f_a^0 f_a^0(\theta_a)$ for $\theta_a \in [\underline{a}^1, \delta_a]$. Moreover, $f_k^1(x)$ is increasing and $f_k^0(x)$ is decreasing over $\mathcal{T}_k$. According to Lemma 4, for each case, function $\Psi_{\mathcal{C}}(\theta_a(t), \theta_b(t))$ is increasing in $\theta_a(t)$ and $\theta_b(t)$.

## H   Proof of Theorem 4

Some notations are simplified by removing subscript $t$ as mentioned in Appendix E.

Note that $f_{k,t}(x) = f_k(x)$ is fixed. Consider two one-shot problems under the same distributions at two consecutive time steps with group representation disparity $\frac{\widetilde{\overline{\alpha}}_a}{\overline{\alpha}_b}$ and $\frac{\widehat{\overline{\alpha}}_a}{\overline{\alpha}_b}$ respectively. Let $(\widetilde{\theta}_a, \widetilde{\theta}_b)$ and $(\widehat{\theta}_a, \widehat{\theta}_b)$ be the corresponding solutions.

According to Lemma 2, $\widetilde{\theta}_a, \widehat{\theta}_a \in [\phi_{\mathcal{C}}(\delta_b), \delta_a], \widetilde{\theta}_b, \widehat{\theta}_b \in [\delta_b, \phi_{\mathcal{C}}^{-1}(\delta_a)]$ hold. Suppose $\frac{\widetilde{\overline{\alpha}}_a(t)}{\overline{\alpha}_b(t)} > \frac{\widehat{\overline{\alpha}}_a}{\overline{\alpha}_b}$. By Theorem 3, it implies that $\widetilde{\theta}_a > \widehat{\theta}_a, \widetilde{\theta}_b > \widehat{\theta}_b$.

Consider the dynamics with $\pi_k(\theta_k) = \nu(L_k(\theta_k))$, since $L_k(\theta_k)$ is decreasing over $[\underline{k}^0, \delta_k]$ and increasing over $[\delta_k, \overline{k}^1]$, the larger one-shot decisions $\theta_a, \theta_b$ would result in the larger retention rate $\pi_a(\theta_a)$ and the smaller $\pi_b(\theta_b)$ as $\nu(\cdot)$ is strictly decreasing. Therefore, $\pi_a(\widetilde{\theta}_a) > \pi_a(\widehat{\theta}_a)$ and $\pi_b(\widetilde{\theta}_b) < \pi_b(\widehat{\theta}_b)$. Hence, Monotonicity condition is satisfied.

Consider the dynamics with $\pi_k(\theta_k) = w(D_k(\theta_k))$ where $D_k(\theta_k) = \int_{\theta_k}^\infty g_k^1 f_k^1(x) - g_k^0 f_k^0(x) dx$. The following holds for $G_a$ and $G_b$:

$$D_a(\theta_a) = \int_{\delta_a}^\infty g_a^1 f_a^1(x) - g_a^0 f_a^0(x) dx + \int_{\theta_a}^{\delta_a} g_a^1 f_a^1(x) - g_a^0 f_a^0(x) dx$$

$$D_b(\theta_b) = \int_{\delta_b}^\infty g_b^1 f_b^1(x) - g_b^0 f_b^0(x) dx - \int_{\delta_b}^{\theta_b} g_b^1 f_b^1(x) - g_b^0 f_b^0(x) dx$$

Since $g_a^1 f_a^1(x) \leq g_a^0 f_a^0(x)$ for $x \leq \delta_a$ and $g_b^1 f_b^1(x) \geq g_b^0 f_b^0(x)$ for $x \geq \delta_b$, the larger $\theta_a$, $\theta_b$ will thus result in the larger $\pi_a(\theta_a)$ and smaller $\pi_b(\theta_b)$ as $w(\cdot)$ is strictly increasing. Therefore, $\pi_a(\widetilde{\theta}_a) > \pi_a(\widehat{\theta}_a)$ and $\pi_b(\widetilde{\theta}_b) < \pi_b(\widehat{\theta}_b)$. Hence, Monotonicity condition is satisfied.

Combine with Theorem 1, $\frac{\overline{\alpha}_a(t)}{\overline{\alpha}_a(t)}$ changes monotonically. By Theorem 3, the corresponding one-shot fair decision $(\theta_a(t), \theta_b(t))$ also converges monotonically.

# I  Proof of Theorem 5

## I.1  Lemmas

To begin, we first introduce some lemmas for two cases. Lemma 5 and 7 show that under the same group representation $\overline{\alpha}_a, \overline{\alpha}_b$, the impact of reshaping distributions on the resulting one-shot decisions. Lemma 6 and 8 demonstrate a sufficient condition on feature distributions and one-shot decisions of two problems such that their expected losses satisfy certain conditions. The proof of these lemmas are presented in Appendix J.

**Case (i):** $f_{k,t}(x) = g_{k,t}^1 f_k^1(x) + g_{k,t}^0 f_k^0(x)$:

Fraction of subgroup $G_k^j$ over $G_k$ changes according to change of their own perceived loss $L_k^j$, i.e., for $i \in \{0, 1\}$ such that $L_{k,t}^i(\theta_k(t)) < L_{k,t-1}^i(\theta_k(t-1))$, $g_{k,t}^i > g_{k,t-1}^i$ and $g_{k,t}^{-i} < g_{k,t-1}^{-i}$.

**Lemma 5.** *Let $(\widehat{\theta}_a, \widehat{\theta}_b)$, $(\widetilde{\theta}_a, \widetilde{\theta}_b)$ be two pairs of decisions under any of EqOpt, StatPar, Simple fairness criteria such that $\widehat{\Psi}_C(\widehat{\theta}_a, \widehat{\theta}_b) = \widetilde{\Psi}_C(\widetilde{\theta}_a, \widetilde{\theta}_b)$, where functions $\widehat{\Psi}_C$, $\widetilde{\Psi}_C$ have the form given in Table 1 and are defined under feature distributions $\widehat{f}_k(x) = \widehat{g}_k^1 f_k^1(x) + \widehat{g}_k^0 f_k^0(x)$, $\widetilde{f}_k(x) = \widetilde{g}_k^1 f_k^1(x) + \widetilde{g}_k^0 f_k^0(x)$ respectively $\forall k \in \{a, b\}$. If $\widehat{g}_k^1 < \widetilde{g}_k^1$ and $\widehat{g}_k^0 > \widetilde{g}_k^0$, then $\widehat{\theta}_k > \widetilde{\theta}_k$ will hold $\forall k \in \{a, b\}$.*

**Lemma 6.** *Consider two one-shot problems defined in (1) with objectives $\widetilde{O}(\theta_a, \theta_b; \widetilde{\overline{\alpha}}_a, \widetilde{\overline{\alpha}}_b)$ and $\widehat{O}(\theta_a, \theta_b; \widehat{\overline{\alpha}}_a, \widehat{\overline{\alpha}}_b)$, where $\widetilde{O}$ is defined over distributions $\widetilde{f}_k(x) = \widetilde{g}_k^0 f_k^0(x) + \widetilde{g}_k^1 f_k^1(x)$ and $\widehat{O}$ is defined over distributions $\widehat{f}_k(x) = \widehat{g}_k^0 f_k^0(x) + \widehat{g}_k^1 f_k^1(x)$, $k \in \{a, b\}$. Let $(\widetilde{\theta}_a, \widetilde{\theta}_b)$, $(\widehat{\theta}_a, \widehat{\theta}_b)$ be the corresponding one-shot decisions under any of Simple, EqOpt or StatPar fairness criteria. For any $\widehat{g}_k^0 + \widehat{g}_k^1 = 1$ and $\widetilde{g}_k^0 + \widetilde{g}_k^1 = 1$ such that $\widehat{g}_k^0 > \widetilde{g}_k^0$, $\widehat{g}_k^1 < \widetilde{g}_k^1$, $\forall k \in \{a, b\}$, if $\widehat{\theta}_a > \widetilde{\theta}_a$ and $\widehat{\theta}_b > \widetilde{\theta}_b$, then $\widehat{L}_a(\widehat{\theta}_a) < \widetilde{L}_a(\widetilde{\theta}_a)$ and $\widehat{L}_b(\widehat{\theta}_b) > \widetilde{L}_b(\widetilde{\theta}_b)$ can be satisfied under the following condition:*

$$|\Delta g_k(\widetilde{L}_k^0(\widetilde{\theta}_k) - \widetilde{L}_k^1(\widetilde{\theta}_k))| < |\int_{\widetilde{\theta}_k}^{\widehat{\theta}_k} \widehat{g}_k^0 f_k^0(x) - \widehat{g}_k^1 f_k^1(x) dx|, \ \forall k \in \{a, b\} \tag{7}$$

*where $\Delta g_k = |\widehat{g}_k^0 - \widetilde{g}_k^0| = |\widehat{g}_k^1 - \widetilde{g}_k^1|$.*

Note that Condition (7) can be satisfied when: (1) $\Delta g_k$ is sufficiently small; and (2) the difference in the decision $\widehat{\theta}_k - \widetilde{\theta}_k$ is sufficiently large, which can be achieved if $\widehat{\overline{\alpha}}_k$ and $\widetilde{\overline{\alpha}}_k$ are quite different.

**Case (ii):** $f_{k,t}(x) = g_k^1 f_{k,t}^1(x) + g_k^0 f_{k,t}^0(x)$

Suppose $L_{k,t}^1(\theta_k(t)) > L_{k,t-1}^1(\theta_k(t-1))$, i.e., $G_k^1$ is less and less favored by the decision over time, then users from $G_k^1$ will make additional effort to improve their features so that $f_{k,t}^1(x)$ will skew toward the direction of higher feature value, i.e., $f_{k,t+1}^1(x) < f_{k,t}^1(x)$ for $x$ with smaller value ($x \in \mathcal{T}_k$) while $G_k^0$ is assumed to be unaffected, i.e., $f_{k,t+1}^0(x) = f_{k,t}^0(x)$. Similar statements hold

when $\theta_k(t) < \theta_k(t-1)$ and $G_k^0$ is less and less favored. Moreover, assume that Assumption 1 holds for any reshaped distributions and the support of $f_{k,t}^1(x)$ and $f_{k,t}^0(x)$ do not change over time.

$\forall t$, let $f_{k,t}^0(x)$ and $f_{k,t}^1(x)$ overlap over $\mathcal{T}_k := [\underline{k}^1, \overline{k}^0]$.

**Lemma 7.** *Let $(\widehat{\theta}_a, \widehat{\theta}_b)$, $(\widetilde{\theta}_a, \widetilde{\theta}_b)$ be two pairs of decisions under any of EqOpt, StatPar, Simple fairness criteria such that $\widehat{\Psi}_C(\widehat{\theta}_a, \widehat{\theta}_b) = \widetilde{\Psi}_C(\widetilde{\theta}_a, \widetilde{\theta}_b)$, where functions $\widehat{\Psi}_C$, $\widetilde{\Psi}_C$ have the form given in Table 1 and are defined under feature distributions $\widehat{f}_k(x) = g_k^1 \widehat{f}_k^1(x) + g_k^0 \widehat{f}_k^0(x)$, $\widetilde{f}_k(x) = g_k^1 \widetilde{f}_k^1(x) + g_k^0 \widetilde{f}_k^0(x)$ respectively $\forall k \in \{a, b\}$. If $\widehat{f}_k^0(x) = \widetilde{f}_k^0(x)$ and $\widehat{f}_k^1(x) < \widetilde{f}_k^1(x)$, $\forall x \in \mathcal{T}_k$, then $\widehat{\theta}_k > \widetilde{\theta}_k$ will hold $\forall k \in \{a, b\}$.*

**Lemma 8.** *Consider two one-shot problems defined in (1) with objectives $\widetilde{O}(\theta_a, \theta_b; \widetilde{\widehat{\alpha}}_a, \widetilde{\widehat{\alpha}}_b)$ and $\widehat{O}(\theta_a, \theta_b; \widehat{\widehat{\alpha}}_a, \widehat{\widehat{\alpha}}_b)$, where $\widetilde{O}$ is defined over distributions $\widetilde{f}_k(x) = g_k^0 \widetilde{f}_k^0(x) + g_k^1 \widetilde{f}_k^1(x)$ and $\widehat{O}$ is defined over distributions $\widehat{f}_k(x) = g_k^0 \widehat{f}_k^0(x) + g_k^1 \widehat{f}_k^1(x)$, $k \in \{a, b\}$. Let $(\widetilde{\theta}_a, \widetilde{\theta}_b)$, $(\widehat{\theta}_a, \widehat{\theta}_b)$ be the corresponding one-shot decisions under any of Simple, EqOpt or StatPar fairness criteria. For any distributions $\widetilde{f}_k^1$, $\widehat{f}_k^1$ increasing over $\mathcal{T}_k$ and $\widetilde{f}_k^0$, $\widehat{f}_k^0$ decreasing over $\mathcal{T}_k$ such that $\widehat{f}_k^1(x) < \widetilde{f}_k^1(x)$ over $\mathcal{T}_k$ and $\widehat{f}_k^0(x) = \widetilde{f}_k^0(x) = f_k^0(x), \forall x, \forall k \in \{a, b\}$. if $\widehat{\theta}_a > \widetilde{\theta}_a$ and $\widehat{\theta}_b > \widetilde{\theta}_b$, then $\widehat{L}_a(\widehat{\theta}_a) < \widetilde{L}_a(\widetilde{\theta}_a)$ holds. Moreover, $\widehat{L}_b(\widehat{\theta}_b) > \widetilde{L}_b(\widetilde{\theta}_b)$ can be satisfied under the following condition:*

$$\Delta f_b^1 g_b^1 (\max\{\widetilde{\theta}_b, \widehat{\delta}_b\} - \underline{b}^1) < \int_{\max\{\widetilde{\theta}_b, \widehat{\delta}_b\}}^{\widehat{\theta}_b} g_b^1 \widehat{f}_b^1(x) - g_b^0 \widehat{f}_b^0(x) dx \tag{8}$$

*where $\Delta f_b^1 = \max_{x \in [\underline{b}^1, \max\{\widetilde{\theta}_b, \widehat{\delta}_b\}]} |\widehat{f}_b^1(x) - \widetilde{f}_b^1(x)|$ and $\widehat{\delta}_b$ is defined such that $g_b^0 \widehat{f}_b^0(\widehat{\delta}_b) = g_b^1 \widehat{f}_b^1(\widehat{\delta}_b)$.*

Note that Condition (8) can be satisfied when: (1) $\Delta f_b^1$ is sufficiently small, which makes $\widehat{\delta}_b$ close to $\widetilde{\delta}_b$ and $\widetilde{\theta}_b = \max\{\widetilde{\theta}_b, \widehat{\delta}_b\}$ is more likely to hold; and (2) the difference in the decision $\widehat{\theta}_b - \widetilde{\theta}_b$ is sufficiently large, which can be achieved if $\widehat{\widehat{\alpha}}_k$ and $\widetilde{\widehat{\alpha}}_k$ are quite different.

## I.2 Sufficient conditions

Below we formally state the sufficient condition under which Theorem 5 can hold.

**Condition 1.** *[Sufficient condition for exacerbation] Condition 1 is satisfied if the following holds:*

- *under **Case (i)**: Condition (7) is satisfied for objectives $O_t$ and $O_{t+1}$, $\forall t \geq 2$, i.e.,*

$$|\Delta g_{k,t+1}(L_{k,t}^0(\theta_k^r(t)) - L_{k,t}^1(\theta_k^r(t)))| < |\int_{\theta_k^r(t)}^{\theta_k^r(t+1)} g_{k,t+1}^0 f_k^0(x) - g_{k,t+1}^1 f_k^1(x) dx|, k \in \{a, b\}$$

  *with $\Delta g_{k,t+1} = |g_{k,t+1}^j - g_{k,t}^j|, j \in \{0, 1\}$.*

- *under **Case (ii)**: Condition (8) is satisfied for objectives $O_t$ and $O_{t+1}$, $\forall t \geq 2$, i.e.,*

$$\Delta f_{b,t+1}^1 g_b^1 (\max\{\theta_b^r(t), \delta_{b,t+1}\} - \underline{b}^1) < \int_{\max\{\theta_b^r(t), \delta_{b,t+1}\}}^{\theta_b^r(t+1)} g_b^1 f_{b,t+1}^1(x) - g_b^0 f_{b,t+1}^0(x) dx$$

  *with $\Delta f_{b,t+1}^1 = \max_{x \in [\underline{b}^1, \max\{\theta_b^r(t), \delta_{b,t+1}\}]} |f_{b,t+1}^1(x) - f_{b,t}^1(x)|$.*

**Condition 2.** *[Sufficient condition for acceleration of exacerbation]*

*Let $O_t^f := O_t^f(\theta_a, \theta_b; \overline{\alpha}_a^f(t), \overline{\alpha}_b^f(t))$ be the objective of the one-shot problem at time $t$ for the case when distributions are fixed over time. Condition 2 is satisfied if the following holds:*

- *under **Case (i)**: Condition (7) is satisfied for objectives $O_t$ and $O_t^f$, $\forall t \geq 2$, i.e.,*

$$|\Delta g_{k,t}(L_{k,t}^0(\theta_k^f(t)) - L_{k,t}^1(\theta_k^f(t)))| < |\int_{\theta_k^f(t)}^{\theta_k^r(t)} g_{k,t}^0 f_k^0(x) - g_{k,t}^1 f_k^1(x) dx|, k \in \{a, b\}$$

  *with $\Delta g_{k,t} = g_{k,t}^j - g_{k,1}^j, j \in \{0, 1\}$.*

- *under **Case (ii)**: Condition (8) is satisfied for objectives $\boldsymbol{O}_t$ and $\boldsymbol{O}_t^f$, $\forall t \geq 2$, i.e.,*

$$\Delta f_{b,t}^1 g_b^1 (\max\{\theta_b^f(t), \delta_{b,t}\} - \underline{b}^1) < \int_{\max\{\theta_b^f(t),\delta_{b,t}\}}^{\theta_b^r(t)} g_b^1 f_{b,t}^1(x) - g_b^0 f_{b,t}^0(x) dx$$

$$\text{with } \Delta f_{b,t}^1 = \max_{x \in [\underline{b}^1, \max\{\theta_b^f(t), \delta_{b,t}\}]} |f_{b,t}^1(x) - f_{b,1}^1(x)|.$$

Note that Condition 1 is likely to be satisfied when changing the decision from $\theta_k(t)$ to $\theta_k(t+1)$ results in: (i) a minor change of $f_{k,t+1}(x)$ from $f_{k,t}(x)$; or/and (ii) a significant change of representation disparity $\frac{\overline{\alpha}_a(t+1)}{\overline{\alpha}_b(t+1)}$ from $\frac{\overline{\alpha}_a(t)}{\overline{\alpha}_b(t)}$ so that $|\theta_k^r(t+1) - \theta_k^r(t)|$ is sufficiently large.

Condition 2 is likely to be satisfied if for any time step, (i) the change of $f_{k,t}(x)$ is minor as compared to the fixed distribution, i.e., $f_{k,1}(x)$ at time $t = 1$; or/and (ii) the resulting decisions at same time under two schemes are quite different, i.e., $|\theta_k^f(t) - \theta_k^r(t)|$ is sufficiently large.

In other words, both requires that $f_{k,t}(x)$ is relatively insensitive to the change of one-shot decisions, and this applies to scenarios where the impact of reshaping distributions is considered as a slow process, e.g., change of credit score takes time and is a slow process.

## I.3  Proof of main theorem

If $f_{k,t}(x) = f_k(x)$ is fixed $\forall t$, then the relationship between $\frac{\overline{\alpha}_a^f(t)}{\overline{\alpha}_b^f(t)}$ and one-shot solutions $(\theta_a^f(t), \theta_b^f(t))$ follows $\frac{\overline{\alpha}_a^f(t)}{\overline{\alpha}_b^f(t)} = \Psi_{C,1}(\theta_a^f(t), \theta_b^f(t)), \forall t$. If $f_{k,t}(x)$ varies over time, then $\frac{\overline{\alpha}_a^r(t)}{\overline{\alpha}_b^r(t)} = \Psi_{C,t}(\theta_a^r(t), \theta_b^r(t)), \forall t$. We consider that distributions start to change after individuals feel the change of perceived decisions, i.e., $f_{k,t}(x)$ begins to change at time $t = 3$. In the following $\forall k \in \{a, b\}$, $\theta_k^f(t) = \theta_k^r(t) = \theta_k(t)$, $\pi_{k,t}^f(\theta_k^f(t)) = \pi_{k,t}^r(\theta_k^r(t)) = \pi_{k,t}(\theta_k(t))$ for $t = 1, 2$ and $\frac{\overline{\alpha}_a^f(t)}{\overline{\alpha}_b^f(t)} = \frac{\overline{\alpha}_a^r(t)}{\overline{\alpha}_b^r(t)} = \frac{\overline{\alpha}_a(t)}{\overline{\alpha}_b(t)}$ for $t = 1, 2, 3$.

Start from $t = 1$, if $(\theta_a(1), \theta_b(1))$ satisfies $\pi_{a,1}(\theta_a(1)) > \pi_{b,1}(\theta_b(1))$, then $\frac{\overline{\alpha}_a(2)}{\overline{\alpha}_b(2)} > \frac{\overline{\alpha}_a(1)}{\overline{\alpha}_b(1)}$ and $\theta_k(2) > \theta_k(1)$ holds $\forall k \in \{a, b\}$, implying $\pi_{a,2}(\theta_a(2)) > \pi_{a,1}(\theta_a(1)) > \pi_{b,1}(\theta_b(1)) > \pi_{b,2}(\theta_b(2))$ ($\boldsymbol{O}_1$ and $\boldsymbol{O}_2$ satisfy monotonicity condition) and $\frac{\overline{\alpha}_a(3)}{\overline{\alpha}_b(3)} > \frac{\overline{\alpha}_a(2)}{\overline{\alpha}_b(2)}$. Moreover, the change of decisions begins to reshape the feature distributions in the next time step.

Consider two ways of reshaping distributions: **Case (i)** and **Case (ii)**. For both cases, show that as long as the change of distribution from $f_{k,t-1}(x)$ to $f_{k,t}(x)$ is relatively small w.r.t. the change of decision from $\theta_k(t-2)$ to $\theta_k(t-1)$ (formally stated in Condition 1 and Condition 2), the following can hold for any time step $t \geq 3$: (i) $\boldsymbol{O}_t$ and $\boldsymbol{O}_{t+1}$ satisfy monotonicity condition: $\pi_{a,t+1}^r(\theta_a^r(t+1)) > \pi_{a,t}^r(\theta_a^r(t))$, $\pi_{b,t}^r(\theta_b^r(t)) > \pi_{b,t+1}^r(\theta_b^r(t+1))$ hold when $\frac{\overline{\alpha}_a^r(t+1)}{\overline{\alpha}_b^r(t+1)} > \frac{\overline{\alpha}_a^r(t)}{\overline{\alpha}_b^r(t)}$; (ii) group representation disparity changes faster than case when distributions are fixed, i.e., $\frac{\overline{\alpha}_a^r(t)}{\overline{\alpha}_b^r(t)} \geq \frac{\overline{\alpha}_a^f(t)}{\overline{\alpha}_b^f(t)}, \forall t$.

Since $\theta_k(2) > \theta_k(1)$, within the same group $G_k$, subgroup $G_k^1$ (resp. $G_k^0$) experiences the higher (resp. lower) loss at time $t = 2$ than $t = 1$. Consider two types of change $\forall k \in \{a, b\}$:

- **Case (i)**: $g_{k,3}^1 < g_{k,2}^1 = g_{k,1}^1$ and $g_{k,3}^0 > g_{k,2}^0 = g_{k,1}^0$.

- **Case (ii)**: $f_{k,3}^0(x) = f_{k,2}^0(x) = f_{k,1}^0(x), \forall x$ and $f_{k,3}^1(x) < f_{k,2}^1(x) = f_{k,1}^1(x), \forall x \in \mathcal{T}_k$.

Prove the following by induction under Condition 1 and 2 (on the sensitivity of $f_{k,t}(x)$ w.r.t. the change of decisions): For $t > 3$, $\frac{\overline{\alpha}_a^r(t+1)}{\overline{\alpha}_b^r(t+1)} > \frac{\overline{\alpha}_a^f(t+1)}{\overline{\alpha}_b^f(t+1)}$ and $\frac{\overline{\alpha}_a^r(t+1)}{\overline{\alpha}_b^r(t+1)} > \frac{\overline{\alpha}_a^r(t)}{\overline{\alpha}_b^r(t)}$ hold, and $\forall k \in \{a, b\}$:

- **Case (i)**: $g_{k,t+1}^1 < g_{k,t}^1 < g_{k,1}^1$ and $g_{k,t+1}^0 > g_{k,t}^0 > g_{k,1}^0$ are satisfied.

- **Case (ii)**: $f_{k,t+1}^0(x) = f_{k,t}^0(x) = f_{k,1}^0(x), \forall x$ and $f_{k,t+1}^1(x) < f_{k,t}^1(x) < f_{k,1}^1(x), \forall x \in \mathcal{T}_k$.

**Base case:**

$\Psi_{C,t}$ are defined under feature distributions $f_{k,t}(x) = g_{k,t}^1 f_{k,t}^1(x) + g_{k,t}^0 f_{k,t}^0(x), \forall k \in \{a, b\}$. Define a pair $(\tilde{\theta}_a, \tilde{\theta}_b)$ such that the following holds:

$\frac{\overline{\alpha}_a(3)}{\overline{\alpha}_b(3)} = \Psi_{C,1}(\theta_a^f(3), \theta_b^f(3)) = \Psi_{C,3}(\theta_a^r(3), \theta_b^r(3)) = \Psi_{C,2}(\tilde{\theta}_a, \tilde{\theta}_b) > \Psi_{C,2}(\theta_a^r(2), \theta_b^r(2)) = \frac{\overline{\alpha}_a(2)}{\overline{\alpha}_b(2)}.$

Then, we have $\forall k \in \{a, b\}$:

• **Case (i)**: As $g_{k,3}^1 < g_{k,2}^1 = g_{k,1}^1$ and $g_{k,3}^0 > g_{k,2}^0 = g_{k,1}^0$, by Lemma 5, $\theta_k^r(3) > \theta_k^f(3) = \tilde{\theta}_k$ holds.

• **Case (ii)**: As $f_{k,3}^0(x) = f_{k,2}^0(x) = f_{k,1}^0(x), \forall x$ and $f_{k,3}^1(x) < f_{k,2}^1(x) < f_{k,1}^1(x), \forall x \in \mathcal{T}_k$, by Lemma 7, $\theta_k^r(3) > \theta_k^f(3) = \tilde{\theta}_k$ holds.

By Theorem 3, $\tilde{\theta}_k > \theta_k^r(2)$ holds. It implies that $\theta_k^r(3) > \theta_k^f(3)$ and $\theta_k^r(3) > \theta_k^r(2)$.

Consider dynamics with $\pi_{k,t}(\theta_k(t)) = \nu(L_{k,t}(\theta_k(t)))$. The following statements hold:

(1) Under Condition 1, $L_{a,3}(\theta_a^r(3)) < L_{a,2}(\theta_a^r(2))$ and $L_{b,3}(\theta_b^r(3)) > L_{b,2}(\theta_b^r(2))$ hold, implying $\pi_{a,3}^r(\theta_a^r(3)) > \pi_{a,2}^r(\theta_a^r(2)) > \pi_{b,2}^r(\theta_b^r(2)) > \pi_{b,3}^r(\theta_b^r(3))$ and $\frac{\overline{\alpha}_a^r(4)}{\overline{\alpha}_b^r(4)} > \frac{\overline{\alpha}_a^r(3)}{\overline{\alpha}_b^r(3)}$.

(2) Under Condition 2, $L_{a,3}(\theta_a^r(3)) < L_{a,3}(\theta_a^f(3))$ and $L_{b,3}(\theta_b^r(3)) > L_{b,3}(\theta_b^f(3))$ hold, implying $\pi_{a,3}^r(\theta_a^r(3)) > \pi_{a,3}^f(\theta_a^f(3)) > \pi_{b,3}^f(\theta_b^f(3)) > \pi_{b,3}^r(\theta_b^r(3))$ and $\frac{\overline{\alpha}_a^r(4)}{\overline{\alpha}_b^r(4)} > \frac{\overline{\alpha}_a^f(4)}{\overline{\alpha}_b^f(4)}$.

(3) $G_k^1$ (resp. $G_k^0$) experiences the higher (resp. lower) loss at $t = 3$ than $t = 2$, i.e., $L_{k,3}^1(\theta_k^r(3)) > L_{k,2}^1(\theta_k^r(2))$ and $L_{k,3}^0(\theta_k^r(3)) < L_{k,2}^0(\theta_k^r(2))$,

• **Case (i)**: $g_{k,4}^1 < g_{k,3}^1 < g_{k,1}^1$ and $g_{k,4}^0 > g_{k,3}^0 > g_{k,1}^0$ hold.

• **Case (ii)**: $f_{k,4}^0(x) = f_{k,3}^0(x) = f_{k,1}^0(x), \forall x$ and $f_{k,4}^1(x) < f_{k,3}^1(x) < f_{k,1}^1(x), \forall x \in \mathcal{T}_k$ hold.

**Induction step:**

Suppose at time $t > 3$, $\frac{\overline{\alpha}_a^r(t+1)}{\overline{\alpha}_b^r(t+1)} > \frac{\overline{\alpha}_a^f(t+1)}{\overline{\alpha}_b^f(t+1)}$ and $\frac{\overline{\alpha}_a^r(t+1)}{\overline{\alpha}_b^r(t+1)} > \frac{\overline{\alpha}_a^r(t)}{\overline{\alpha}_b^r(t)}$ hold, and $\forall k \in \{a, b\}$:

• **Case (i)**: $g_{k,t+1}^1 < g_{k,t}^1 < g_{k,1}^1$ and $g_{k,t+1}^0 > g_{k,t}^0 > g_{k,1}^0$ are satisfied.

• **Case (ii)**: $f_{k,t+1}^0(x) = f_{k,t}^0(x) = f_{k,1}^0(x), \forall x$ and $f_{k,t+1}^1(x) < f_{k,t}^1(x) < f_{k,1}^1(x), \forall x \in \mathcal{T}_k$.

Then consider time step $t + 1$.

Define pairs $(\tilde{\theta}_a, \tilde{\theta}_b)$ and $(\hat{\theta}_a, \hat{\theta}_b)$ such that the following holds:

$$\frac{\overline{\alpha}_a^r(t+1)}{\overline{\alpha}_b^r(t+1)} = \Psi_{C,t+1}(\theta_a^r(t+1), \theta_b^r(t+1)) > \begin{cases} \frac{\overline{\alpha}_a^f(t+1)}{\overline{\alpha}_b^f(t+1)} = \Psi_{C,1}(\theta_a^f(t+1), \theta_b^f(t+1)) = \Psi_{C,t+1}(\tilde{\theta}_a, \tilde{\theta}_b) \\ \frac{\overline{\alpha}_a^r(t)}{\overline{\alpha}_b^r(t)} = \Psi_{C,t}(\theta_a^r(t), \theta_b^r(t)) = \Psi_{C,t+1}(\hat{\theta}_a, \hat{\theta}_b) \end{cases}$$

According to the hypothesis, Under **Case (i)**, $\tilde{\theta}_k > \theta_k^f(t+1)$ and $\hat{\theta}_k > \theta_k^r(t)$ hold by Lemma 5. Under **Case (ii)**, $\tilde{\theta}_k > \theta_k^f(t+1)$ and $\hat{\theta}_k > \theta_k^r(t)$ hold by Lemma 7. By Theorem 3, $\theta_k^r(t+1) > \tilde{\theta}_k$ and $\theta_k^r(t+1) > \hat{\theta}_k$ hold. It implies that $\theta_k^r(t+1) > \theta_k^f(t+1)$ and $\theta_k^r(t+1) > \theta_k^r(t)$.

(1) Under Condition 1, $L_{a,t+1}(\theta_a^r(t+1)) < L_{a,t}(\theta_a^r(t))$ and $L_{b,t+1}(\theta_b^r(t+1)) > L_{b,t}(\theta_b^r(t))$ hold, implying $\pi_{a,t+1}^r(\theta_a^r(t+1)) > \pi_{a,t}^r(\theta_a^r(t)) > \pi_{b,t}^r(\theta_b^r(t)) > \pi_{b,t+1}^r(\theta_b^r(t+1))$ and $\frac{\overline{\alpha}_a^r(t+1)}{\overline{\alpha}_b^r(t+1)} > \frac{\overline{\alpha}_a^r(t)}{\overline{\alpha}_b^r(t)}$: $\mathbf{O}_t$ and $\mathbf{O}_{t+1}$ satisfy monotonicity condition and representation disparity get exacerbated.

(2) Under Condition 2, $L_{a,t+1}(\theta_a^r(t+1)) < L_{a,t+1}(\theta_a^f(t+1))$ and $L_{b,t+1}(\theta_b^r(t+1)) > L_{b,t+1}(\theta_b^f(t+1))$ hold, implying $\pi_{a,t+1}^r(\theta_a^r(t+1)) > \pi_{a,t+1}^f(\theta_a^f(t+1)) > \pi_{b,t+1}^f(\theta_b^f(t+1)) > \pi_{b,t+1}^r(\theta_b^r(t+1))$ and thus $\frac{\overline{\alpha}_a^r(t+1)}{\overline{\alpha}_b^r(t+1)} > \frac{\overline{\alpha}_a^f(t+1)}{\overline{\alpha}_b^f(t+1)}$: the discrepancy between retention rates of two demographic groups is larger at each time compared to the case when distributions are fixed, and if the disparity get exacerbated, this exacerbation is accelerated under the reshaping.

(3) $G_k^1$ (resp. $G_k^0$) experiences the higher (resp. lower) loss at $t + 1$ than $t$, i.e., $L_{k,t+1}^1(\theta_k^r(t+1)) > L_{k,t}^1(\theta_k^r(t))$ and $L_{k,t+1}^0(\theta_k^r(t+1)) < L_{k,t}^0(\theta_k^r(t))$. Therefore,

• **Case (i)**: $g_{k,t+2}^1 < g_{k,t+1}^1 < g_{k,1}^1$ and $g_{k,t+2}^0 > g_{k,t+1}^0 > g_{k,1}^0$ hold.

• **Case (ii)**: $f_{k,t+2}^0(x) = f_{k,t+1}^0(x) = f_{k,1}^0(x), \forall x$ and $f_{k,t+2}^1(x) < f_{k,t+1}^1(x) < f_{k,1}^1(x), \forall x \in \mathcal{T}_k$ hold.

Proof is completed.

The case if $\pi_{a,1}(\theta_a(1)) < \pi_{b,1}(\theta_b(1))$ can be proved similarly and is omitted.

# J    Proof of Lemmas for Theorem 5

## J.1    Proof of Lemma 5

$f_k^0(x)$ and $f_k^1(x)$ overlap over $\mathcal{T}_k := [\underline{k}^1, \overline{k}^0]$.

1. $\mathcal{C} := \texttt{StatPar}$

To satisfy $\widehat{\Psi}_{\texttt{StatPar}}(\widehat{\theta}_a, \widehat{\theta}_b) = \widetilde{\Psi}_{\texttt{StatPar}}(\widetilde{\theta}_a, \widetilde{\theta}_b)$, $\frac{\widehat{g}_k^1 f_k^1(\widehat{\theta}_k)}{\widehat{g}_k^0 f_k^0(\widehat{\theta}_k)} = \frac{\widetilde{g}_k^1 f_k^1(\widetilde{\theta}_k)}{\widetilde{g}_k^0 f_k^0(\widetilde{\theta}_k)}$ should hold. Under Assumption 1, both $\frac{\widehat{g}_k^1 f_k^1(\cdot)}{\widehat{g}_k^0 f_k^0(\cdot)}$ and $\frac{\widetilde{g}_k^1 f_k^1(\cdot)}{\widetilde{g}_k^0 f_k^0(\cdot)}$ are strictly increasing over $\mathcal{T}_k$. Since $\forall k \in \{a, b\}$, there is $\frac{\widehat{g}_k^1 f_k^1(\theta_k)}{\widehat{g}_k^0 f_k^0(\theta_k)} < \frac{\widetilde{g}_k^1 f_k^1(\theta_k)}{\widetilde{g}_k^0 f_k^0(\theta_k)}$, $\forall \theta_k \in \mathcal{T}_k$. For all three possibilities in Table 1, $\widehat{\theta}_k > \widetilde{\theta}_k$ holds $\forall k \in \{a, b\}$.

2. $\mathcal{C} := \texttt{EqOpt}$

Since $\widetilde{L}_a^0(\theta_a) = \widetilde{L}_b^0(\theta_b)$ and $\widehat{L}_a^0(\theta_a) = \widehat{L}_b^0(\theta_b)$ always hold for any $(\theta_a, \theta_b)$ satisfying $\texttt{EqOpt}$ criterion, when change of $\widehat{g}_k^0$ (or $\widetilde{g}_k^0$) is determined by $\theta_k$ only via $\widehat{L}_k^0(\theta_k)$ (or $\widetilde{L}_k^0(\theta_k)$), both $\frac{\widehat{g}_b^0}{\widehat{g}_a^0} = 1$ and $\frac{\widetilde{g}_b^0}{\widetilde{g}_a^0} = 1$ are satisfied. To satisfy $\widehat{\Psi}_{\texttt{EqOpt}}(\widehat{\theta}_a, \widehat{\theta}_b) = \widetilde{\Psi}_{\texttt{EqOpt}}(\widetilde{\theta}_a, \widetilde{\theta}_b)$, $\frac{\widehat{g}_k^1 f_k^1(\widehat{\theta}_k)}{\widehat{g}_k^0 f_k^0(\widehat{\theta}_k)} = \frac{\widetilde{g}_k^1 f_k^1(\widetilde{\theta}_k)}{\widetilde{g}_k^0 f_k^0(\widetilde{\theta}_k)}$ should hold, which is same as the condition that should be satisfied in case when $\mathcal{C} := \texttt{StatPar}$. Rest of the proof is thus same as $\texttt{StatPar}$ case and is omitted.

3. $\mathcal{C} := \texttt{Simple}$

$\texttt{Simple}$ fairness criterion requires that $\widehat{\theta}_a = \widehat{\theta}_b = \widehat{\theta}$ and $\widetilde{\theta}_a = \widetilde{\theta}_b = \widetilde{\theta}$. In order to satisfy $\widehat{\Psi}_{\texttt{Simple}}(\widehat{\theta}_a, \widehat{\theta}_b) = \widetilde{\Psi}_{\texttt{Simple}}(\widetilde{\theta}_a, \widetilde{\theta}_b)$, $\frac{\widehat{g}_b^1 f_b^1(\widehat{\theta}) - \widehat{g}_b^0 f_b^0(\widehat{\theta})}{\widehat{g}_a^0 f_a^0(\widehat{\theta}) - \widehat{g}_a^1 f_a^1(\widehat{\theta})} = \frac{\widetilde{g}_b^1 f_b^1(\widetilde{\theta}) - \widetilde{g}_b^0 f_b^0(\widetilde{\theta})}{\widetilde{g}_a^0 f_a^0(\widetilde{\theta}) - \widetilde{g}_a^1 f_a^1(\widetilde{\theta})}$ should hold. Under Assumption 1, both $\frac{\widehat{g}_b^1 f_b^1(\cdot) - \widehat{g}_b^0 f_b^0(\cdot)}{\widehat{g}_a^0 f_a^0(\cdot) - \widehat{g}_a^1 f_a^1(\cdot)}$ and $\frac{\widetilde{g}_b^1 f_b^1(\cdot) - \widetilde{g}_b^0 f_b^0(\cdot)}{\widetilde{g}_a^0 f_a^0(\cdot) - \widetilde{g}_a^1 f_a^1(\cdot)}$ are strictly increasing over $\mathcal{T}_k$. Since $\forall k \in \{a, b\}$, there is $\frac{\widehat{g}_b^1 f_b^1(\theta) - \widehat{g}_b^0 f_b^0(\theta)}{\widehat{g}_a^0 f_a^0(\theta) - \widehat{g}_a^1 f_a^1(\theta)} < \frac{\widetilde{g}_b^1 f_b^1(\theta) - \widetilde{g}_b^0 f_b^0(\theta)}{\widetilde{g}_a^0 f_a^0(\theta) - \widetilde{g}_a^1 f_a^1(\theta)}$, $\forall \theta \in \mathcal{T}_a \cap \mathcal{T}_b$, implying that $\widehat{\theta} > \widetilde{\theta}$.

## J.2    Proof of Lemma 6

Define $\Delta L_k^j = |\widehat{L}_k^j(\widehat{\theta}_k) - \widetilde{L}_k^j(\widetilde{\theta}_k)|, j \in \{0, 1\}$. Rewrite $\widehat{g}_k^0 = \widetilde{g}_k^0 + \Delta g_k$ and $\widehat{g}_k^1 = \widetilde{g}_k^1 - \Delta g_k$. For $k \in \{a, b\}$, $\widehat{\theta}_k > \widetilde{\theta}_k$ holds, which implies that $\widehat{L}_k^1(\widehat{\theta}_k) = \widetilde{L}_k^1(\widetilde{\theta}_k) + \Delta L_k^1$ and $\widehat{L}_k^0(\widehat{\theta}_k) = \widetilde{L}_k^0(\widetilde{\theta}_k) - \Delta L_k^0$. Therefore,

$$\widehat{L}_k(\widehat{\theta}_k) - \widetilde{L}_k(\widetilde{\theta}_k) = \Delta g_k(\widetilde{L}_k^0(\widetilde{\theta}_k) - \widetilde{L}_k^1(\widetilde{\theta}_k)) - (\widehat{g}_k^0 \Delta L_k^0 - \widehat{g}_k^1 \Delta L_k^1), k \in \{a, b\}$$

since

$$\Delta L_k^1 = \int_{\widetilde{\theta}_k}^{\widehat{\theta}_k} f_k^1(x) dx; \ \Delta L_k^0 = \int_{\widetilde{\theta}_k}^{\widehat{\theta}_k} f_k^0(x) dx$$

Define $\widehat{\delta}_k$ such that $\widehat{g}_k^0 f_k^0(\widehat{\delta}_k) = \widehat{g}_k^1 f_k^1(\widehat{\delta}_k)$, then $\widehat{g}_a^0 f_a^0(x) > \widehat{g}_a^1 f_a^1(x)$ when $x < \widehat{\delta}_a$ and $\widehat{g}_b^0 f_b^0(x) < \widehat{g}_b^1 f_b^1(x)$ when $x > \widehat{\delta}_b$. By Lemma 2, $\widehat{\theta}_a < \widehat{\delta}_a$ and $\widehat{\theta}_b > \widehat{\delta}_b$ hold, implying

$$\widehat{g}_k^0 \Delta L_k^0 - \widehat{g}_k^1 \Delta L_k^1 = \int_{\widetilde{\theta}_k}^{\widehat{\theta}_k} \widehat{g}_k^0 f_k^0(x) - \widehat{g}_k^1 f_k^1(x) dx \begin{cases} > 0, & k = a \\ < 0, & k = b \end{cases}$$

If $|\Delta g_k(\widetilde{L}_k^0(\widetilde{\theta}_k) - \widetilde{L}_k^1(\widetilde{\theta}_k))| < |\int_{\widetilde{\theta}_k}^{\widehat{\theta}_k} \widehat{g}_k^0 f_k^0(x) - \widehat{g}_k^1 f_k^1(x) dx|$ holds, then the sign of $\widehat{L}_k(\widehat{\theta}_k) - \widetilde{L}_k(\widetilde{\theta}_k)$ is determined by the sign of $\widehat{g}_k^1 \Delta L_k^1 - \widehat{g}_k^0 \Delta L_k^0$. We have $\widehat{L}_a(\widehat{\theta}_a) < \widetilde{L}_a(\widetilde{\theta}_a)$ and $\widehat{L}_b(\widehat{\theta}_b) > \widetilde{L}_b(\widetilde{\theta}_b)$.

## J.3 Proof of Lemma 7

1. $\mathcal{C} := \texttt{StatPar}$ or $\mathcal{C} := \texttt{EqOpt}$

To satisfy $\widehat{\Psi}_{\texttt{StatPar}}(\widehat{\theta}_a, \widehat{\theta}_b) = \widetilde{\Psi}_{\texttt{StatPar}}(\widetilde{\theta}_a, \widetilde{\theta}_b)$ or $\widehat{\Psi}_{\texttt{EqOpt}}(\widehat{\theta}_a, \widehat{\theta}_b) = \widetilde{\Psi}_{\texttt{EqOpt}}(\widetilde{\theta}_a, \widetilde{\theta}_b)$, $\frac{g_k^1 \widetilde{f}_k^1(\widetilde{\theta}_k)}{g_k^0 \widetilde{f}_k^0(\widetilde{\theta}_k)} = \frac{g_k^1 \widehat{f}_k^1(\widehat{\theta}_k)}{g_k^0 \widehat{f}_k^0(\widehat{\theta}_k)} < \frac{g_k^1 \widetilde{f}_k^1(\widehat{\theta}_k)}{g_k^0 \widehat{f}_k^0(\widehat{\theta}_k)}$ should hold. Under Assumption 1, $\frac{g_k^1 \widehat{f}_k^1(\cdot)}{g_k^0 \widehat{f}_k^0(\cdot)}$ is strictly increasing over $\mathcal{T}_k$. $\widehat{\theta}_k > \widetilde{\theta}_k$ has to be satisfied.

2. $\mathcal{C} := \texttt{Simple}$

Simple fairness criterion requires that $\widehat{\theta}_a = \widehat{\theta}_b = \widehat{\theta}$ and $\widetilde{\theta}_a = \widetilde{\theta}_b = \widetilde{\theta}$. In order to satisfy $\widehat{\Psi}_{\texttt{Simple}}(\widehat{\theta}_a, \widehat{\theta}_b) = \widetilde{\Psi}_{\texttt{Simple}}(\widetilde{\theta}_a, \widetilde{\theta}_b)$, $\frac{g_b^1 \widehat{f}_b^1(\widehat{\theta}) - g_b^0 \widehat{f}_b^0(\widehat{\theta})}{g_a^0 \widehat{f}_a^0(\widehat{\theta}) - g_a^1 \widehat{f}_a^1(\widehat{\theta})} = \frac{g_b^1 \widehat{f}_b^1(\widehat{\theta}) - g_b^0 \widehat{f}_b^0(\widehat{\theta})}{g_a^0 \widehat{f}_a^0(\widehat{\theta}) - g_a^1 \widehat{f}_a^1(\widehat{\theta})} < \frac{g_b^1 \widehat{f}_b^1(\widehat{\theta}) - g_b^0 \widehat{f}_b^0(\widehat{\theta})}{g_a^0 \widehat{f}_a^0(\widehat{\theta}) - g_a^1 \widehat{f}_a^1(\widehat{\theta})}$ should hold. Under Assumption 1, $\frac{g_b^1 \widehat{f}_b^1(\cdot) - g_b^0 \widehat{f}_b^0(\cdot)}{g_a^0 \widehat{f}_a^0(\cdot) - g_a^1 \widehat{f}_a^1(\cdot)}$ is strictly increasing over $\mathcal{T}_k$. For $\widehat{\theta}, \widetilde{\theta} \in \mathcal{T}_a \cap \mathcal{T}_b$, $\widehat{\theta} > \widetilde{\theta}$ has to be satisfied.

## J.4 Proof of Lemma 8

Define $\widehat{\delta}_k$ such that $g_k^0 \widehat{f}_k^0(\widehat{\delta}_k) = g_k^1 \widehat{f}_k^1(\widehat{\delta}_k)$. Then, $g_a^0 \widehat{f}_a^0(x) > g_a^1 \widehat{f}_a^1(x)$ when $x < \widehat{\delta}_a$ and $g_b^0 \widehat{f}_b^0(x) < g_b^1 \widehat{f}_b^1(x)$ when $x > \widehat{\delta}_b$.

Since $\widehat{\theta}_k > \widetilde{\theta}_k$, we have

$$\widehat{L}_k^0(\widehat{\theta}_k) - \widetilde{L}_k^0(\widetilde{\theta}_k) = -\int_{\widetilde{\theta}_k}^{\widehat{\theta}_k} \widetilde{f}_k^0(x) dx = -\int_{\widetilde{\theta}_k}^{\widehat{\theta}_k} \widehat{f}_k^0(x) dx$$

$$\widehat{L}_k^1(\widehat{\theta}_k) - \widetilde{L}_k^1(\widetilde{\theta}_k) = \int_{\widetilde{\theta}_k}^{\widehat{\theta}_k} \widehat{f}_k^1(x) dx - \int_{\underline{k}^1}^{\widetilde{\theta}_k} (\widetilde{f}_k^1(x) - \widehat{f}_k^1(x)) dx$$

Therefore,

$$\widehat{L}_k(\widehat{\theta}_k) - \widetilde{L}_k(\widetilde{\theta}_k) = \int_{\widetilde{\theta}_k}^{\widehat{\theta}_k} g_k^1 \widehat{f}_k^1(x) - g_k^0 \widehat{f}_k^0(x) dx - g_k^1 \int_{\underline{k}^1}^{\widetilde{\theta}_k} (\widetilde{f}_k^1(x) - \widehat{f}_k^1(x)) dx$$

since $\widetilde{\theta}_a < \widehat{\theta}_a < \widehat{\delta}_a$, $\int_{\widetilde{\theta}_a}^{\widehat{\theta}_a} g_a^1 \widehat{f}_a^1(x) - g_a^0 \widehat{f}_a^0(x) dx < 0$ holds. Since $\widetilde{f}_a^1(x) > \widehat{f}_a^1(x)$ for $x \in \mathcal{T}_a$, we have $g_a^1 \int_{\underline{a}^1}^{\widetilde{\theta}_a} (\widetilde{f}_a^1(x) - \widehat{f}_a^1(x)) dx > 0$. Therefore, $\widehat{L}_a(\widehat{\theta}_a) < \widetilde{L}_a(\widetilde{\theta}_a)$.

When $k = b$, there are two possibilities: (i) $\widetilde{\theta}_b < \widehat{\delta}_b < \widehat{\theta}_b$; (ii) $\widehat{\delta}_b < \widetilde{\theta}_b < \widehat{\theta}_b$.

For case (i),

$$\widehat{L}_b(\widehat{\theta}_b) - \widetilde{L}_b(\widetilde{\theta}_b) = \underbrace{\int_{\widehat{\delta}_b}^{\widehat{\theta}_b} g_b^1 \widehat{f}_b^1(x) - g_b^0 \widehat{f}_b^0(x) dx}_{\textbf{term 1}} + \underbrace{\int_{\widetilde{\theta}_b}^{\widehat{\delta}_b} g_b^1 \widehat{f}_b^1(x) - g_b^0 \widehat{f}_b^0(x) dx}_{\textbf{term 2}}$$

$$+ \underbrace{g_b^1 \int_{\underline{b}^1}^{\widetilde{\theta}_b} (\widehat{f}_b^1(x) - \widetilde{f}_b^1(x)) dx}_{\textbf{term 3}}$$

Since $\widehat{\delta}_b < \widetilde{\theta}_b < \widehat{\delta}_b$ and $\widehat{f}_b^0(x) = \widetilde{f}_b^0(x)$, for $x \in [\widetilde{\theta}_b, \widehat{\delta}_b]$, $g_b^1 \widehat{f}_b^1(x) - g_b^1 \widetilde{f}_b^1(x) < g_b^1 \widehat{f}_b^1(x) - g_b^0 \widehat{f}_b^0(x) < 0$, we have $0 > \textbf{term 2} + \textbf{term 3} > g_b^1 \int_{\underline{b}^1}^{\widehat{\delta}_b} (\widehat{f}_b^1(x) - \widetilde{f}_b^1(x)) dx$.

Define $\Delta_1 = \max_{x \in [\underline{b}^1, \widehat{\delta}_b]} |\widehat{f}_b^1(x) - \widetilde{f}_b^1(x)|$. Since $\textbf{term 1} > 0$, $\widehat{L}_b(\widehat{\theta}_b) > \widetilde{L}_b(\widetilde{\theta}_b)$ holds only if the following condition is satisfied:

$$\Delta_1 g_b^1 (\widehat{\delta}_b - \underline{b}^1) < \int_{\widehat{\delta}_b}^{\widehat{\theta}_b} g_b^1 \widehat{f}_b^1(x) - g_b^0 \widehat{f}_b^0(x) dx$$

For case (ii),

$$\widehat{L}_b(\widehat{\theta}_b) - \widetilde{L}_b(\widetilde{\theta}_b) = \underbrace{\int_{\widetilde{\theta}_b}^{\widehat{\theta}_b} g_b^1 \widehat{f}_b^1(x) - g_b^0 \widehat{f}_b^0(x)dx}_{\textbf{term 1}} + \underbrace{g_b^1 \int_{\underline{b}^1}^{\widetilde{\theta}_b} (\widehat{f}_b^1(x) - \widetilde{f}_b^1(x))dx}_{\textbf{term 2}}$$

Define $\Delta_2 = \max_{x \in [\underline{b}^1, \widetilde{\theta}_b]} |\widehat{f}_b^1(x) - \widetilde{f}_b^1(x)|$. Similar to case (i), $\widehat{L}_b(\widehat{\theta}_b) > \widetilde{L}_b(\widetilde{\theta}_b)$ holds only if the following condition is satisfied:

$$\Delta_2 g_b^1 (\widetilde{\theta}_b - \underline{b}^1) < \int_{\widetilde{\theta}_b}^{\widehat{\theta}_b} g_b^1 \widehat{f}_b^1(x) - g_b^0 \widehat{f}_b^0(x)dx$$

Combine two cases, let $\Delta f_b^1 = \max_{x \in [\underline{b}^1, \max\{\widetilde{\theta}_b, \widehat{\delta}_b\}]} |\widehat{f}_b^1(x) - \widetilde{f}_b^1(x)|$, $\widehat{L}_b(\widehat{\theta}_b) > \widetilde{L}_b(\widetilde{\theta}_b)$ holds only if the following condition is satisfied:

$$\Delta f_b^1 g_b^1 (\max\{\widetilde{\theta}_b, \widehat{\delta}_b\} - \underline{b}^1) < \int_{\max\{\widetilde{\theta}_b, \widehat{\delta}_b\}}^{\widehat{\theta}_b} g_b^1 \widehat{f}_b^1(x) - g_b^0 \widehat{f}_b^0(x)dx$$

## K  More on examples of finding proper fairness constraints from dynamics

**Example 1.** *[Linear first order model] is given by* $N_k(t+1) = N_k(t)\pi_k^2(\theta_k(t)) + \beta_k \pi_k^1(\theta_k(t))$. *This is a general form of dynamics* (2) *where the arrivals can also depend on the decision. When* $\pi_k^1(\theta_k(t)) = 1$, *then dynamics model will be reduced to* (2). $\tilde{N}_k = \frac{\beta_k \pi_k^1(\theta_k)}{1 - \pi_k^2(\theta_k)}$ *is the stable fixed point if* $\pi_k^2(\theta_k) < 1$ *holds. Since* $|\frac{\tilde{N}_a}{\tilde{N}_b} - \frac{\beta_a}{\beta_b}| = \frac{\beta_a}{\beta_b}|\frac{\pi_a^1(\theta_a)}{\pi_b^1(\theta_b)}\frac{1 - \pi_b^2(\theta_b)}{1 - \pi_a^2(\theta_a)} - 1|$, *solution pair* $(\theta_a^*, \theta_b^*)$ *should satisfy* $\frac{\pi_a^1(\theta_a^*)}{1 - \pi_a^2(\theta_a^*)} = \frac{\pi_b^1(\theta_b^*)}{1 - \pi_b^2(\theta_b^*)}$. *The constraint set that can sustain the group representation is given by:*

$$\mathcal{C} = \{(\theta_a, \theta_b) | (\theta_a, \theta_b) \in \Theta \times \Theta, \frac{\pi_a^1(\theta_a)}{1 - \pi_a^2(\theta_a)} = \frac{\pi_b^1(\theta_b)}{1 - \pi_b^2(\theta_b)}, \pi_a^2(\theta_a) < 1, \pi_b^2(\theta_b) < 1\}.$$

Consider the case where departure is driven by positive rate $\pi_k^2(\theta_k) = \nu(\int_{\theta_k}^{\infty} f_k(x)dx)$ and arrival is driven by error rate $\pi_k^1(\theta_k) = \nu(g_k^0 \int_{\theta_k}^{\infty} f_k^0(x)dx + g_k^1 \int_{-\infty}^{\theta_k} f_k^1(x)dx) = \nu(L_k(\theta_k))$ where $\nu(\cdot)$ is a strictly decreasing function. This can be applied in lending scenario, where an applicant will stay as long as he/she gets the loan (positive rate) regardless of his/her qualification. Since an unqualified applicant who is issued the loan cannot repay, his/her credit score will be decreased which lowers the chance to get a loan in the future [13]. Therefore, users may decide whether to apply for a loan based on the error rate.

In Fig. 5, $\Delta$-fair set is illustrated for the case when $f_k^j(x)$, $k \in \{a, b\}, j \in \{0, 1\}$ is truncated normal distributed with parameters $[\sigma_a^0, \sigma_a^1, \sigma_b^0, \sigma_b^1] = [5, 6, 6, 5]$, $[\underline{k}^0, \underline{k}^1, \overline{k}^0, \overline{k}^1] = [5, 11, 20, 35]$, $[\mu_k^0, \mu_k^1] = [10, 25]$ for $k \in \{a, b\}$ and $\nu(x) = 1 - x$. The left heat map illustrates the $\Delta$-fair set for the dynamics model mentioned above. On the other hand, the right heat map illustrates the dynamics model introduced in Section 3.2 where the departure is driven by model accuracy, i.e., $\pi_k^2(\theta_k) = \nu(L_k(\theta_k))$ and $\pi_k^1(\theta_k) = 1$. Here, $x$-axis and $y$-axis represent $\theta_b$ and $\theta_a$ respectively. Each pair $(\theta_a, \theta_b)$ has a corresponding value of $|\frac{\tilde{N}_a}{\tilde{N}_b} - \frac{\beta_a}{\beta_b}|$ measuring how well it can sustain the group representation. The colored area illustrates all the pairs such that $|\frac{\tilde{N}_a}{\tilde{N}_b} - \frac{\beta_a}{\beta_b}| \leq \frac{\beta_a}{\beta_b}$. All $(\theta_a, \theta_b)$ pairs that have the same value of $|\frac{\tilde{N}_a}{\tilde{N}_b} - \frac{\beta_a}{\beta_b}| = \frac{\beta_a}{\beta_b}\epsilon$ form a curve of the same color, where the corresponding value of $\epsilon \in [0, 1]$ is shown in the color bar. $\Delta$-fair set is the union of all curves with $\epsilon \leq \Delta\frac{\beta_b}{\beta_a}$.

**Example 2.** *[Quadratic first order model] is given by* $N_k(t+1) = (N_k(t))^2 \pi_k^1(\theta_k(t)) + \beta_k$. $\tilde{N}_k = \frac{1}{2\pi_k^1(\theta_k)} - \sqrt{\frac{1}{4(\pi_k^1(\theta_k))^2} - \frac{\beta_k}{\pi_k^1(\theta_k)}}$ *is the stable fixed point if* $\pi_k^1(\theta_a) < \frac{1}{4\beta_k}$ *holds. Since* $|\frac{\tilde{N}_a}{\tilde{N}_b} - \frac{\beta_a}{\beta_b}| = \frac{\beta_a}{\beta_b}|\frac{\beta_b \pi_b^1(\theta_b)}{\beta_a \pi_a^1(\theta_a)}\frac{1 - \sqrt{1 - 4\beta_a \pi_a^1(\theta_a)}}{1 - \sqrt{1 - 4\beta_b \pi_b^1(\theta_b)}} - 1|$, *then* $\beta_a \pi_a^1(\theta_a^*) = \beta_b \pi_b^1(\theta_b^*)$ *should be satisfied. The constraint set that can sustain the group representation is given by*

$$\mathcal{C} = \{(\theta_a, \theta_b) | (\theta_a, \theta_b) \in \Theta \times \Theta, \beta_a \pi_a^1(\theta_a) = \beta_b \pi_b^1(\theta_b), \pi_a^1(\theta_a) < \frac{1}{4\beta_a}, \pi_b^1(\theta_b) < \frac{1}{4\beta_b}\}.$$

# L    Supplementary Material for the Experiments

## L.1    Parameter settings

$f_k^j(x)$ follows the truncated normal distribution, the supports of $f_k^j(x), k \in \{a,b\}, j \in \{0,1\}$ are $[\underline{a}^0, \underline{a}^1, \overline{a}^0, \overline{a}^1] = [-8, 5, 19, 35], [\underline{b}^0, \underline{b}^1, \overline{b}^0, \overline{b}^1] = [-6, 25, 9, 43]$, with the means $[\mu_a^0, \mu_a^1, \mu_b^0, \mu_b^1] = [4, 20, 8, 27]$ and standard deviations $[\sigma_a^0, \sigma_a^1, \sigma_b^0, \sigma_b^1] = [5, 6, 3, 6]$. The label proportions are $g_a^0 = 0.4$, $g_b^0 = 0.6$. The dynamics (2) uses $\nu(x) = 1 - x$.

## L.2    Illustration of convergence of sample paths

(a) Group proportion          (b) Average total loss

Fig. 9: Sample paths for truncated normal example under different fairness criteria when $\beta_a + \beta_b = 20000$. Group proportion $\overline{\alpha}_a(t)$ and average total loss are shown in Fig.9(a)9(b) respectively: solid lines are for the case $\beta_a = \beta_b$, dashed lines for $\beta_a = 3\beta_b$, and dotted dashed lines for $\beta_a = \beta_b/3$.

Fig. 9 shows sample paths of the group proportion and average total loss using one-shot fair decisions and different combinations of $\beta_a, \beta_b$ under dynamics with $\pi_{k,t}(\cdot) = \nu(L_{k,t}(\cdot))$. In all cases convergence is reached (we did not include the decisions $\theta_k(t)$ but convergence holds there as well). In particular, under EqLos fairness, the group representation is sustained throughout the horizon. By contrast, under other fairness constraints, even a "major" group (one with a larger arrival $\beta_k$) can be significantly marginalized over time (blue/green dashed line in Fig. 9(a)). This occurs when the loss of the minor group happens to be smaller than that of the major group, which is determined by feature distributions of the two groups (see Fig. 10). Whenever this is the case, the one-shot fair decision will seek to increase the minor group's proportion in order to drive down the average loss.

(a) Feature distributions illustration          (b) Group proportion $\beta_a = \beta_b$

Fig. 10: Change $f_b^0(x)$ by varying $\sigma_b^0 \in \{1, 2, 3, 4, 5, 6, 7\}$. As $\sigma_b^0$ increases, the overlap area with $f_b^1(x)$ also increases as shown in Fig. 10(a). Fig. 10(b) shows the result under StatPar fairness. Given $\theta_a(t)$, the larger $\sigma_b^0$ results in the larger $L_b(\theta_b(t))$ and thus the smaller $G_b$'s retention rate.

## L.3    Dynamics driven by other factors

To sustain the group representation, the key point is that the fairness definition should match the factors that drive user departure and arrival. If adopt different dynamic models, different fairness

criteria should be adopted. Two examples with different dynamics and the performance of four fairness criteria are demonstrated in Fig. 11.

(a) Users from $G_k$ are driven by false negative rate

(b) Users from $G_k^j$ are driven by their own perceived loss

Fig. 11: Sample paths under different dynamic models: Three cases are demonstrated including $\beta_a = \beta_b$ (solid curves); $\beta_a = 3\beta_b$ (dashed curves); $\beta_a = \beta_b/3$ (dotted dash curves). Fig. 11(a) illustrates the model where the user departure is driven by false negative rate: $N_k(t+1) = N_k(t)\nu(\text{FN}_k(\theta_k(t))) + \beta_k$, with $\text{FN}_k(\theta_k(t)) = \int_{\theta_k(t)}^{\infty} f_k^0(x)dx$. Under this model EqOpt is better at maintaining representation. Fig. 11(b) illustrates the model where the users from each sub-group $G_k^j$ are driven by their own perceived loss: $N_k^j(t+1) = N_k^j(t)\nu(L_k^j(\theta_k(t))) + g_k^j \beta_k$, with $L_k^j(\theta_k)$ being false positives for $j = 0$ and false negatives for $j = 1$. Under this model none of the four criteria can maintain group representation.

## L.4 When distributions are learned from users in the system

If $f_k^j(x)$ is unknown to the decision maker and the decision is learned from users in the system, then as users leave the system the decision can be more inaccurate and the exacerbation could potentially get more severe. In order to illustrate this, we first modify the dynamic model such that the users' arrivals are also effected by the model accuracy,[11] i.e., $N_k(t+1) = (N_k(t) + \beta_k)\nu(L_k(\theta_k(t)))$. We compare the performance of two cases: *(i)* the Bayes optimal decisions are applied in every round; and *(ii)* decisions in $(t+1)$th round are learned from the remaining users in $t$th round. The empirical results are shown in Fig. 12 where each solid curve (resp. dashed curve) is a sample path of case *(i)* (resp. case *(ii)*). Although $\beta_a = \beta_b$, $G_b$ suffers a smaller loss at the beginning and starts to dominate the overall objective gradually. It results in the less and less users from group $G_a$ than $G_b$ in the sample pool and the model trained from minority group $G_a$ suffers an additional loss due to its insufficient samples. In contrast, as $G_b$ dominates more in the objective and its loss may be decreased compared with the case *(i)* (See Fig. 12(c)). As a consequence, the exacerbation in group representation disparity gets more severe (See Fig. 12(a)).

(a) Group proportion

(b) $G_a$'s total population

(c) $G_b$'s total population

Fig. 12: Impact of the classifier's quality: dashed curves represent the results for decisions learned from users (case *(ii)*), solid curves represent the results for Bayes optimal decisions (case *(i)*). It shows the exacerbation of group disparity get more severe under case *(ii)* for Simple, EqOpt and StatPar criteria.