[Reviews · NeurIPS 2019]

Reviewer 1



Originality: To the best of my knowledge the model of general user retention dynamics and corresponding statements evidencing negative feedback loops are novel contributions to the literature in sequential fairness works. The contributions of the paper would be clearer if citations were provided for methods and models introduced in earlier works (for example, I suggest adding citations for the fairness criteria in lines 149-158, for user departure models in 197-208, and for the statement in lines 173-174, if applicable). Since the full related work is deferred to the appendix, I see no need to cite [2, 3, 7, 10, 15, 16] without distinction between them. More context on what these works do and how they relate to your work is useful for readers to contextualize your contributions; please expand on the discussion of these papers. Quality: The simple and unifying model of sequential decision making presented is very valuable in my opinion. The assumptions necessary for the technical results (lemmas and theorems) are stated clearly. The assumptions regarding monotonicity and group separation seem fairly restrictive, but are technically illuminating if not practical for real-data regimes. The statement in lines 173-174 regrading strictly increasing function \phi_{C,t} either needs to be backed up mathematically with reference to a proof, or with a citation. Clarity: I could not find forward links to any of the appendix proofs in the main text. Please provide references to the appendix where applicable so that the interested reader can find them easily. Additionally, since the paper is very notation heavy, it would be nice to provide a link to the notation table at the front of the appendix. The figures are highly informative, and add to the exposition and understanding, but are unfortunately not conducive to the page size (figures are too small to be read without significant zooming). Specifically: - Figures 1, 2, and 3 are too small to be read in a printed version. Please either make the figure sizes bigger or make sure that the quality of the images is high-resolution and the text within figures is big enough that it can be read on a printed - The color scale for figures 4 and 5 needs to be a sequential color scale. As it currently is, the plots are not intelligible in a black and white print out since the orange, green, and magenta all correspond to the same grey-scale. I'm not sure that Table 1 is the most effective way to show the group representation disparity changes, as the expressions themselves are quite hard to parse. Perhaps there is a more interpretable way to summarize this set of results, for example by showing properties of the resulting functions, rather than the functions themselves, in the table. Parentheses in Table 1 should be bigger for the fractions, i.e. use \left( and \right) Overall the prose is well written in the main text. I suggest you revise the appendix for grammar, particularly in the mathematically heavy sections (some sentences did not parse). Purely to aid in your revisions, here's a list of specific lines at which I found typos/grammatical mistakes/etc.: 18, 39, 118 (missing space), 167-168, 176, 367, 764. Significance: While the assumptions are not necessarily indicative of real-data regimes, they are theoretically illuminating, and clearly stated. I believe that this is a good starting point for future work which could extend the results presented here to characterize regimes in which one could hope to say, incentivize decisions that in the long term equalize representation rates. The definition and analysis of simple models here is a valuable building block for the community. However, the presentation of these results is slightly hindered by the sheer amount of notation and technical results without major discussion. Figures are not readable at regular scale, and little to no intuition for proofs is given in the main text (while no forward links to the appendix are provided for proofs). Lastly, little is said with regards to the feasibility and applicability of the proposed model in different real-world contexts. The paper, and I believe it's ultimate impact, would be strengthened by a discussion of what the proposed framework adds to scholarship on sequential decisions in the fairness literature, beyond the results presented here. Therefore, in addition to points raised above, I request the authors to address the following more open-ended questions in their response: - When aiming to promote stable group representations via the proportion of each group in the total population, what ensures that the absolute sizes of both groups are not driven to be very small (\beta_a, \beta_b), even if the ratios become more even? Is this a scenario that is considered bad in your framework? - Regarding the experimental results in Section 4, is the case of EqLos special, or do you believe that other dynamics models for group retention will correspond to the 'right' notion of fairness being an already defined fairness constraint?

Reviewer 2



Update: I have read the other reviewers' comments and the author feedback. I update my score to 5. A few more comments: 1. Since the authors did not provide the result (for multi-dim) that they claimed, I am not able to comment on the veracity or significance of such a result. In general, I believe that one can find some setting where MC holds. The question is whether these conditions are reasonable. 2. Using the adult dataset: I'm not sure what the relevant application is here. Why would there be any dynamics at all in this setting? This is more of a proof of concept. I don't think my concerns about the lack of pertinent application domains were fully addressed by this experiment. 4. Hashimoto et al.'s Corollary 1 gave general conditions for when the "fair" fixed point is not stable, which directly implies that the dynamics will converge to some fixed point where there is representation disparity. In other words, this already implies the authors' results (e.g. Theorem 3). Therefore the main technical contribution of this work seems to be to show that "static fairness criteria" does not always help with representation disparity. ==== Originality: User retention model is mostly taken from previous work, i.e. Hashimoto et al 2018. One add is considering "User departure driven by intra-group disparity" but this is given quite little justification apart from lines 204-205, which does not explain why this is a model researchers and practitioners should care about. The idea that representation disparity worsens over time is also introduced in Hashimoto et al 2018 with a more general model (i.e. not just one dimensional features) so the conclusion of this paper is somewhat unsurprising. The authors also considered applying fairness constraints at each step and worked out the population ratios in this case (Theorem 2), but this result appears to be trivial under their setting and assumptions. It's perhaps unsurprising that applying fairness criteria without knowing the dynamics will not prevent representation disparity from getting worse in general. That being said, showing this in the one-dim setting is a contribution and a natural enough extension of previous work, and the proofs look like a fair amount of work even though the arguments are elementary. One concern is that there was no meaningful comparison made between different fairness criteria and the absence of fairness criteria, since there's only a blanket negative result. I'm not sure if this is an interesting takeaway at all, as you can always build a worst case that is not close to anything in reality. Quality: The notation is cumbersome (e.g. g_k,t^i, f_b,t^1(x)), even though most of the results rely on rather simple ideas, e.g. follows from the monotonicity condition, and does not seem to warrant such heavy notation. Clarity: In the 3.1 binary classification problem, what is y? Its distribution was not given. Again, the heavy notation did not help with clarity. Significance: In terms of significance, a weakness of the paper is that all the conclusions are proved in the one dimensional setting, and it's not clear what their analogues are in the multidimensional setting. The authors claimed (falsely, I think) that "the main conclusion holds for multi-dimensional scenarios, when it's not at all obvious that is the multi-dim analogue for Assumption 1, even (and not to mention Theorem 2). Assumption 1 is very specific to the one dimensional setting. The toy one-dim setting seems very far from any application where one might care about user retention, so it does not make a very strong point. The experiments section seemed lacking. The synthetic experiment in the one-dim setting did not connect the paper's results meaningfully with any relevant application. One question is in what application do we suspect that there is widening user disparity over time? Is there any real data that can be used to illustrate this?

Reviewer 3



This paper provides a framework to study group representation dynamic over time, provides theoretical analysis from the perspective of departure dynamic and in-group representation change. The paper is well-written and most importantly, clearly delivers several key insights in this sequential setting. 1. The paper shows that group representation disparity can easily exacerbate under a monotonicity condition, which means the retention rate goes higher when a group's representation increases. While this observation is not new, the authors further show that the exacerbation cannot be solved under commonly fairness criterion because the group feature distribution still can be affected, which is nontrivial but often ignored in fairness research that focus on static supervised learning tasks. 2. Another key contribution of this paper is to highlight the importance of aligning fairness criteria with what drives user retention. Fig 5 in Section 4 shows the distribution of the converged group proportion under combinations of arriving rates. 3. The authors propose a method of finding proper fairness criteria through a constrained optimization step. They show the ideal stable fixed point in a heatmap, but real experiment results would be more effective, especially in the case when perfect fairness is not feasible.

[Author Response · NeurIPS 2019]

## Reviewer 1:

**1. What ensures the absolute sizes of both groups are not very small.** The absolute size of a group is a function of both arrival and retention rates. If one assumes non-zero arrival at each time as the paper does, then size will not diminish regardless of retention. In particular, if group representation is maintained over time, then $\theta_k(t) = \theta_k, \forall t$ and the group size converges to $\frac{\beta_k}{1-\pi_k(\theta_k)}$. The only way for this to be small is if both arrival $\beta_k$ and retention $\pi_k(\theta_k)$ are near-0. If we allow arrival to be a function of, say model accuracy (Sec 3.4), then arrival indeed may diminish; in this case ensuring representation (as shown in Sec 3.4) can simultaneously help prevent zero arrival.

**2. Is the case of `EqLos` special regarding the experimental results?** No, it is not. It works because the user retention is assumed to be driven by model accuracy in our experiments. As illustrated in Fig. 8(a) in Appendix K.3, if user retention is driven by TPR/FNR (e.g., loan application), `EqOpt` would be the proper fairness notion.

**3. Pros & cons, feasibility & applicability of our framework.** Since human decision making is inherently a sequential (and non-memoryless) process, we feel our framework of examining fairness in such a sequential framework is appropriate for real-world settings. The main limitation of such an approach is that it requires sufficiently accurate models capturing the underlying dynamics (what drives the adoption/abandonment of ML algorithms, etc), which is not always available. We believe there is value in performing long-term experiments to better understand such dynamics.

**4. Can this framework illustrate when positive scenarios can be achieved.** In a sense the current model captures positive feedback, for the majority group: better model performance leads to population growth. Case 2 in Sec 3.3 may be viewed as another positive instance: a group can work to change their distribution in light of perceived bias in the algorithm (and if they manage to break the condition stated therein then they may retain representation).

**5. Improving readability:** We will adjust figures, add forward references, fix typos, and discuss intuition/comparisons.

## Reviewer 2:

**1. Applicability to more general settings.** Our results indeed apply more generally to non-classification problems and/or multi-dimensional features. Thm 1 states that the representation disparity worsens as long as the monotonicity condition (MC) holds; no requirement is imposed on dimensionality or objective function or dynamics. The 1D classification problem is one such case satisfying MC (Thm 3). However, it can be shown rigorously that under certain conditions for $\pi_k(\theta_k) = \nu_k(O_k(\theta_k))$ for some decreasing $\nu_k(\cdot)$, Thm 3 holds when feature vector $X \in \mathbb{R}^d$ and the underlying problem can be other supervised (e.g., regression) and unsupervised learning. We will be happy to add this result.

**2. Experiments with non-synthetic data.** We trained binary classifiers over *Adult* dataset by minimizing empirical loss where features are individual info (sex, race, nationality, etc.) and labels their annual income ($\geq$ \$50k or $<$ \$50k). Since the dataset does not reflect dynamics, we assume it follows (2) with $\pi_k(\theta_k) = \nu(L_k(\theta_k))$. We examine the monotonic convergence of representation disparity under `Simple`, `EqOpt` (equalized false positive/negative cost(FPC/FNC)) and `EqLos`, and consider cases where $G_a$, $G_b$ are distinguished by sex, race and nationality. These results (shown on the right) are consistent with the paper.

**3. Clarifications** (i) Goal of Thm 2 is not to find population ratios but to find one-shot solutions given population ratios; their relation is in Eqn. (3). (ii) $Y \in \{0,1\}$ is label with distribution $Pr(Y = j|K = k) = g_{k,t}^j$ and $y$ is its realization.

**4. Distinction from (Hashimoto et al., 2018) [6].** Worsening of representation disparity is observed via simulation in [6] without using fairness ($\theta_a = \theta_b$), and a min-max fair is used to address this. We show the introduction of (any type of) fairness does not necessarily solve this problem and do so using formal analysis. Other differences include the fact we consider the case when feature distributions are reshaped by the decisions (Sec 3.3) and [6] does not.

## Reviewer 3:

**1. Experiment with proposed fairness constraint selection.** $\Delta = \epsilon \frac{\beta_a}{\beta_b}$-fair set found with method in Sec 3.4 (left plot): each curve represents a sample path under different $\epsilon$ where $(\theta_a(t), \theta_b(t))$ is from a small randomly selected subset of $\Delta$-fair set $\forall t$ (to model the situation where perfect fairness is not feasible) and $\frac{\beta_a}{\beta_b} = 1$. We observe that fairness is always violated at beginning in lower plot. This is because the fairness set is found based on stable fixed points, which only concerns fairness in the long run.

**2. Visualization of decisions shaping feature distribution in Sec 3.3.** The right plot above illustrates how distributions would change from $t$ to $t + 1$, when $G_k^1$ (resp. $G_k^0$) experiences the higher (resp. lower) loss at $t$ than $t - 1$.

[Meta-Review · NeurIPS 2019]

There was some disagreement among the reviewers. In a subsequent discussion, the positive points outweighed the negative ones and I am happy to support acceptance. The topic is important, the approach is reasonable and the empirical contribution, despite some caveats, is convincing. That being said, I ask the authors to incorporate the reviewers' suggestions in the final version of their paper.